# Metric-Free Individual Fairness in Online Learning

**Yahav Bechavod**
Hebrew University
yahav.bechavod@cs.huji.ac.il

**Christopher Jung**
University of Pennsylvania
chrjung@seas.upenn.edu

**Zhiwei Steven Wu**
Carnegie Mellon University
zstevenwu@cmu.edu

## Abstract

We study an online learning problem subject to the constraint of individual fairness, which requires that similar individuals are treated similarly. Unlike prior work on individual fairness, we do not assume the similarity measure among individuals is known, nor do we assume that such measure takes a certain parametric form. Instead, we leverage the existence of an *auditor* who detects fairness violations without enunciating the quantitative measure. In each round, the auditor examines the learner's decisions and attempts to identify a pair of individuals that are treated unfairly by the learner. We provide a general reduction framework that reduces online classification in our model to standard online classification, which allows us to leverage existing online learning algorithms to achieve sub-linear regret and number of fairness violations. Surprisingly, in the stochastic setting where the data are drawn independently from a distribution, we are also able to establish PAC-style fairness and accuracy generalization guarantees (Rothblum and Yona (2018)), despite only having access to a very restricted form of fairness feedback. Our fairness generalization bound qualitatively matches the uniform convergence bound of Rothblum and Yona (2018), while also providing a meaningful accuracy generalization guarantee. Our results resolve an open question by Gillen et al. (2018) by showing that online learning under an unknown individual fairness constraint is possible even without assuming a strong parametric form of the underlying similarity measure.

## 1 Introduction

As machine learning increasingly permeates many critical aspects of society, including education, healthcare, criminal justice, and lending, there is by now a vast literature that studies how to make machine learning algorithms fair (see, e.g., Chouldechova and Roth (2018); Podesta et al. (2014); Corbett-Davies and Goel (2018)). Most of the work in this literature tackles the problem by taking the *statistical group fairness* approach that first fixes a small collection of high-level groups defined by protected attributes (e.g., race or gender) and then asks for approximate parity of some statistic of the predictor, such as positive classification rate or false positive rate, across these groups (see, e.g., Hardt et al. (2016); Chouldechova (2017); Kleinberg et al. (2017); Agarwal et al. (2018)). While notions of group fairness are easy to operationalize, they are aggregate in nature without fairness guarantees for finer subgroups or individuals (Dwork et al., 2012; Hébert-Johnson et al., 2018; Kearns et al., 2018).

In contrast, the *individual fairness* approach aims to address this limitation by asking for explicit fairness criteria at an individual level. In particular, the compelling notion of individual fairness proposed in the seminal work of Dwork et al. (2012) requires that similar people are treated similarly. The original formulation of individual fairness assumes that the algorithm designer has access to

a task-specific fairness metric that captures how similar two individuals are in the context of the specific classification task at hand. In practice, however, such a fairness metric is rarely specified, and the lack of metrics has been a major obstacle for the wide adoption of individual fairness. There has been recent work on learning the fairness metric based on different forms of human feedback. For example, Ilvento (2019) provides an algorithm for learning the metric by presenting human arbiters with queries concerning the distance between individuals, and Gillen et al. (2018) provide an online learning algorithm that can eventually learn a Mahalanobis metric based on identified fairness violations. While these results are encouraging, they are still bound by several limitations. In particular, it might be difficult for humans to enunciate a precise quantitative similarity measure between individuals. Moreover, their similarity measure across individuals may not be consistent with any metric (e.g., it may not satisfy the triangle inequality) and is unlikely to be given by a simple parametric function (e.g., the Mahalanobis metric function).

To tackle these issues, this paper studies *metric-free* online learning algorithms for individual fairness that rely on a weaker form of interactive human feedback and minimal assumptions on the similarity measure across individuals. Similar to the prior work of Gillen et al. (2018), we do not assume a pre-specified metric, but instead assume access to an *auditor*, who observes the learner's decisions over a group of individuals that show up in each round and attempts to identify a fairness violation—a pair of individuals in the group that should have been treated more similarly by the learner. Since the auditor only needs to identify such unfairly treated pairs, there is no need for them to enunciate a quantitative measure – to specify the distance between the identified pairs. Moreover, we do not impose any parametric assumption on the underlying similarity measure, nor do we assume that it is actually a metric since we do not require that similarity measure to satisfy the triangle inequality. Under this model, we provide a general reduction framework that can take any online classification algorithm (without fairness constraint) as a black-box and obtain a learning algorithm that can simultaneously minimize cumulative classification loss and the number of fairness violations. Our results in particular remove many strong assumptions in Gillen et al. (2018), including their parametric assumptions on linear rewards and Mahalanobis distances, and thus answer several questions left open in their work.

## 1.1 Overview of Model and Results

We study an online classification problem: over rounds $t = 1, \ldots, T$, a learner observes a small set of $k$ individuals with their feature vectors $(x_\tau^t)_{\tau=1}^k$ in space $\mathcal{X}$. The learner tries to predict the label $y_k^t \in \{0, 1\}$ of each individual with a "soft" predictor $\pi^t$ that predicts $\pi^t(x_\tau^t) \in [0, 1]$ on each $x_\tau^t$ and incurs classification loss $|\pi^t(x_\tau^t) - y_\tau^t|$. Then an auditor will investigate if the learner has violated the individual fairness constraint on any pair of individuals within this round, that is, if there exists $(\tau_1, \tau_2) \in [k]^2$ such that $|\pi^t(x_{\tau_1}^t) - \pi^t(x_{\tau_2}^t)| > d(x_{\tau_1}^t, x_{\tau_2}^t) + \alpha$, where $d \colon \mathcal{X} \times \mathcal{X} \to \mathbb{R}_+$ is an unknown distance function and $\alpha$ denotes the auditor's tolerance. If this violation has occurred on any number of pairs, the auditor will identify one of such pairs and incur a fairness loss of 1; otherwise, the fairness loss is 0. Then the learner will update the predictive policy based on the observed labels and the received fairness feedback. Under this model, our results include:

**A Reduction from Fair Online Classification to Standard Online Classification.** Our reduction-based algorithm can take any no-regret online (batch) classification learner as a black-box and achieve sub-linear cumulative fairness loss and sub-linear regret on mis-classification loss compared to the most accurate policy that is fair on every round. In particular, our framework can leverage the generic exponential weights method (Freund and Schapire, 1997; Cesa-Bianchi et al., 1997; Arora et al., 2012) and also oracle-efficient methods, including variants of Follow-the-Perturbed-Leader (FTPL) (e.g., Syrgkanis et al. (2016); Suggala and Netrapalli (2019)), that further reduces online learning to standard supervised learning or optimization problems. We instantiate our framework using two online learning algorithms (exponential weights and CONTEXT-FTPL), both of which obtain a $\tilde{O}(\sqrt{T})$ on misclassification regret and cumulative fairness loss.

**Fairness and Accuracy Generalization Guarantees.** While our algorithmic results hold under adversarial arrivals of the individuals, in the stochastic arrivals setting we show that the uniform average policy over time is probably approximate correct and fair (PACF) (Rothblum and Yona, 2018)–that is, the policy is approximately fair on almost all random pairs drawn from the distribution and nearly matches the accuracy gurantee of the best fair policy. In particular, we show that the average policy $\pi^{avg}$ with high probability satisfies $\Pr_{x,x'}[|\pi^{avg}(x) - \pi^{avg}(x')| > \alpha + 1/T^{1/4}] \le O(1/T^{1/4})$,

which qualitatively achieves similar PACF uniform convergence sample complexity as Rothblum and Yona (2018).[1] However, we establish our generalization guarantee through fundamentally different techniques. While their work assumes a fully specified metric and i.i.d. data, the learner in our setting can only access the similarity measure through an auditor's limited fairness violations feedback. The main challenge we need to overcome is that the fairness feedback is inherently adaptive–that is, the auditor only provides feedback for the sequence of deployed policies, which are updated adaptively over rounds. In comparison, a fully known metric allows the learner to evaluate the fairness guarantee of all policies simultaneously. As a result, we cannot rely on their uniform convergence result to bound the fairness generalization error, but instead we leverage a probabilistic argument that relates the learner's regret to the distributional fairness guarantee.

## 2   Related Work

**Solving open problems in Gillen et al. (2018).** The most related work to ours is Gillen et al. (2018), which studies the linear contextual bandit problem subject to individual fairness with an unknown Mahalanobis metric. Similar to our work, they also assume an auditor who can identify fairness violations in each round and provide an online learning algorithm with sublinear regret and a bounded number of fairness violations. Our results resolve two main questions left open in their work. First, we assume a weaker auditor who only identifies a single fairness violation (as opposed to all of the fairness violations in their setting). Second, we remove the strong parametric assumption on the Mahalanobis metric and work with a broad class of similarity functions that need not be metric.

Starting with Joseph et al. (2016), there is a different line of work that studies online learning for individual fairness, but subject to a different notion called meritocratic fairness (Jabbari et al., 2017; Joseph et al., 2018; Kannan et al., 2017). These results present algorithms that are "fair" within each round but again rely on strong realizability assumptions–their fairness guarantee depends on the assumption that the outcome variable of each individual is given by a linear function. Gupta and Kamble (2019) also studies online learning subject to individual fairness but with a known metric. They formulate a one-sided fairness constraint across time, called fairness in hindsight, and provide an algorithm with regret $O(T^{M/(M+1)})$ for some distribution-dependent constant $M$.

Our work is related to several others that aim to enforce individual fairness without a known metric. Ilvento (2019) studies the problem of metric learning by asking human arbiters distance queries. Unlike Ilvento (2019), our algorithm does not explicitly learn the underlying similarity measure and does not require asking auditors numeric queries. The PAC-style fairness generalization bound in our work falls under the framework of *probably approximately metric-fairness* due to Rothblum and Yona (2018). However, their work assumes a pre-specified fairness metric and i.i.d. data from the distribution, while we establish our generalization through a sequence of adaptive fairness violations feedback over time. Kim et al. (2018) study a group-fairness relaxation of individual fairness, which requires that similar subpopulations are treated similarly. They do not assume a pre-specified metric for their offline learning problem, but they do assume a metric oracle that returns numeric distance values on random pairs of individuals. Jung et al. (2019) study an offline learning problem with subjective individual fairness, in which the algorithm tries to elicit subjective fairness feedback from human judges by asking them questions of the form "should this pair of individuals be treated similarly or not?" Their fairness generalization takes a different form, which involves taking averages over both the individuals and human judges. We aim to provide a fairness generalization guarantee that holds for almost all individuals from the population.

## 3   Model and Preliminaries

We define the instance space to be $\mathcal{X}$ and its label space to be $\mathcal{Y}$. Throughout this paper, we will restrict our attention to binary labels, that is $\mathcal{Y} = \{0, 1\}$. We write $\mathcal{H} : \mathcal{X} \to \mathcal{Y}$ to denote the hypothesis class and assume that $\mathcal{H}$ contains a constant hypothesis – i.e. there exists $h$ such that $h(x) = 0$ for all $x \in \mathcal{X}$. Also, we allow for convex combination of hypotheses for the purpose of randomizing the prediction and denote the simplex of hypotheses by $\Delta\mathcal{H}$; we call a randomized hypothesis a *policy*.

Sometimes, we assume the existence of an underlying (but unknown) distribution $\mathcal{D}$ over $(\mathcal{X}, \mathcal{Y})$. For each prediction $\hat{y} \in \mathcal{Y}$ and its true label $y \in \mathcal{Y}$, there is an associated misclassification loss, $\ell(\hat{y}, y) = \mathbb{1}(\hat{y} \neq y)$. For simplicity, we overload the notation and write

$$\ell(\pi(x), y) = (1 - \pi(x)) \cdot y + \pi(x) \cdot (1 - y) = \mathbb{E}_{h \sim \pi} [\ell(h(x), y)].$$

## 3.1 Individual Fairness and Auditor

We want our deployed policy $\pi$ to behave fairly in some manner, and we use the individual fairness definition from Dwork et al. (2012) that asserts that "similar individuals should be treated similarly." We assume that there is some distance function $d : \mathcal{X} \times \mathcal{X} \rightarrow \mathbb{R}_+$ over the instance space $\mathcal{X}$ which captures the distance between individuals in $\mathcal{X}$, although $d$ doesn't have to satisfy the triangle inequality. The only requirement on $d$ is that it is always non-negative and symmetric $d(x, x') = d(x', x)$.

**Definition 3.1** (($\alpha, \beta$)-fairness). *Assume $\alpha, \beta > 0$. A policy $\pi \in \Delta\mathcal{H}$ is said to be $\alpha$-fair on pair $(x, x')$, if $|\pi(x) - \pi(x')| \leq d(x, x') + \alpha$. We say policy $\pi$'s $\alpha$-fairness violation on pair $(x, x')$ is*

$$v_\alpha(\pi, (x, x')) = \max(0, |\pi(x) - \pi(x')| - d(x, x') - \alpha).$$

*A policy is $\pi$ is said to be $(\alpha, \beta)$-fair on distribution $\mathcal{D}$, if*

$$\Pr_{(x,x') \sim \mathcal{D}|_\mathcal{X} \times \mathcal{D}|_\mathcal{X}} [|\pi(x) - \pi(x')| > d(x, x') + \alpha] \leq \beta.$$

*A policy $\pi$ is said to be $\alpha$-fair on set $S \subseteq \mathcal{X}$, if for all $(x, x') \in S^2$, it is $\alpha$-fair.*

Although individual fairness is intuitively sound, individual fairness notion requires the knowledge of the distance function $d$ which is often hard to specify. Therefore, we rely on an auditor $\mathcal{J}$ that can detect instances of $\alpha$-unfairness.

**Definition 3.2** (Auditor $\mathcal{J}$). *An auditor $\mathcal{J}_\alpha$ which can have its own internal state takes in a reference set $S \subseteq \mathcal{X}$ and a policy $\pi$. Then, it outputs $\rho$ which is either null or a pair of indices from the provided reference set to denote that there is some positive $\alpha$-fairness violation for that pair. For some $S = (x_1, \ldots, x_n)$,*

$$\mathcal{J}_\alpha(S, \pi) = \begin{cases} \rho = (\rho_1, \rho_2) & \text{if } \exists \rho_1, \rho_2 \in [n].\pi(x_{\rho_1}) - \pi(x_{\rho_2}) - d(x_{\rho_1}, x_{\rho_2}) - \alpha > 0 \\ null & \text{otherwise} \end{cases}$$

*If there exists multiple pairs with some $\alpha$-violation, the auditor can choose one arbitrarily.*

**Remark 3.3.** *Our assumptions on the auditor are much more relaxed than those of Gillen et al. (2018), which require that the auditor outputs whether the policy is $0$-fair (i.e. with no slack) on all pairs $S^2$ exactly. Furthermore, the auditor in Gillen et al. (2018) can only handle Mahalanobis distances. In our setting, because of the internal state of the auditor, the auditor does not have to be a fixed function but rather can be adaptively changing in each round. Finally, we never rely on the fact the distance function $d$ stays the same throughout rounds, meaning all our results extend to the case where the distance function governing the fairness constraints is changing every round.*

## 3.2 Online Batch Classification

We now describe our online batch classification setting. In each round $t = 1, \ldots, T$, the learner deploys some model $\pi^t \in \Delta\mathcal{H}$. Upon seeing the deployed policy $\pi^t$, the environment chooses a batch of $k$ individuals, $(x_\tau^t, y_\tau^t)_{\tau=1}^k$ and possibly, a pair of individuals from that round on which $\pi^t$ will be responsible for any $\alpha$-fairness violation. For simplicity, we write $\bar{x}^t = (x_\tau^t)_{\tau=1}^k$ and $\bar{y}^t = (y_\tau^t)_{\tau=1}^k$. The strategy $z_{\text{FAIR-BATCH}}^t \in \mathcal{Z}_{\text{FAIR-BATCH}}$ that the environment chooses can be described by $z_{\text{FAIR-BATCH}}^t = (\bar{x}^t, \bar{y}^t) \times \rho^t$, where $\rho^t \in [k]^2 \cup \{null\}$. Often, we omit the subscript and simply write $z^t$. If $\rho^t = (\rho_1^t, \rho_2^t)$, then $\pi^t$ will be responsible for the $\alpha$-fairness violation on the pair $(x_{\rho_1^t}^t, x_{\rho_2^t}^t)$. There are two types of losses that we are interested in: misclassification and fairness loss.

**Definition 3.4** (Misclassification Loss). *The (batch) misclassification loss $Err^2$ is*

$$Err(\pi, z^t) = \sum_{\tau=1}^k \ell(\pi(x_\tau^t), y_\tau^t).$$

| **Algorithm 1:** Online Fair Batch Classification | **Algorithm 2:** Online Batch Classification |
|---|---|
| FAIR-BATCH | BATCH |
| **for** $t = 1, \ldots, T$ **do** | **for** $t = 1, \ldots, T$ **do** |
|    Learner deploys $\pi^t$ |    Learner deploys $\pi^t$ |
|    Environment chooses $(\bar{x}^t, \bar{y}^t)$ |    Environment chooses $z^t = (\bar{x}^t, \bar{y}^t)$ |
|    Environment chooses the pair $\rho^t$ |    Learner incurs misclassification loss |
|    $z^t = (\bar{x}^t, \bar{y}^t) \times \rho^t$ |      $\mathrm{Err}(\pi^t, z^t)$ |
|    Learner incurs misclassfication loss $\mathrm{Err}(\pi^t, z^t)$ | **end** |
|    Learner incurs fairness loss $\mathrm{Unfair}(\pi^t, z^t)$ | |
| **end** | |

Figure 1: Comparison between Online Fair Batch Classification and Online Batch Classification: each is summarized by the interaction between the learner and the environment: $(\Delta\mathcal{H} \times \mathcal{Z}_{\text{FAIR-BATCH}})^T$ and $(\Delta\mathcal{H} \times \mathcal{Z}_{\text{BATCH}})^T$ where $\mathcal{Z}_{\text{FAIR-BATCH}} = \mathcal{X}^k \times \mathcal{Y}^k \times ([k]^2 \cup \{null\})$ and $\mathcal{Z}_{\text{BATCH}} = \mathcal{X}^k \times \mathcal{Y}^k$.

**Definition 3.5** (Fairness Loss). *The $\alpha$-fairness loss $\mathrm{Unfair}_\alpha$ is*

$$\mathit{Unfair}_\alpha(\pi, z^t) = \begin{cases} \mathbb{1}\left(\pi(x_{\rho_1^t}^t) - \pi(x_{\rho_2^t}^t) - d(x_{\rho_1^t}^t, x_{\rho_2^t}^t) - \alpha > 0\right) & \textit{if } \rho^t = (\rho_1^t, \rho_2^t) \\ 0 & \textit{otherwise} \end{cases}$$

We want the total misclassification and fairness loss over $T$ rounds to be as small as any $\pi^* \in Q$ for some competitor set $Q$, which we describe now. As said above, each round's reference set, a set of pairs for which the deployed policy will possibly be responsible in terms of $\alpha$-fairness, will be defined in terms of the instances that arrive within that round $\bar{x}^t$. The baseline $Q_\alpha$ that we compete against will be all policies that are $\alpha$-fair on $\bar{x}^t$ for all $t \in [T]$:

$$Q_\alpha = \{\pi \in \Delta\mathcal{H} : \pi \text{ is } \alpha\text{-fair on } \bar{x}^t \text{ for all } t \in [T]\}$$

Note that because $\mathcal{H}$ contains a constant hypothesis which must be 0-fair on all instances, $Q_\alpha$ cannot be empty. The difference in total loss between our algorithm and a fixed $\pi^*$ is called 'regret', which we formally define below.

**Definition 3.6** (Algorithm $\mathcal{A}$). *An algorithm $\mathcal{A} : (\Delta\mathcal{H} \times \mathcal{Z})^* \to \Delta\mathcal{H}$ takes in its past history $(\pi^\tau, z^\tau)_{\tau=1}^{t-1}$ and deploys a policy $\pi^t \in \Delta\mathcal{H}$ at every round $t \in [T]$.*

**Definition 3.7** (Regret). *For some $Q \subseteq \Delta\mathcal{H}$, the regret of algorithm $\mathcal{A}$ with respect to some loss $L : \Delta\mathcal{H} \times \mathcal{Z} \to \mathbb{R}$ is denoted as $\mathbf{Regret}^L(\mathcal{A}, Q, T)$, if for any $(z_t)_{t=1}^T$,*

$$\sum_{t=1}^T L\left(\pi^t, z^t\right) - \inf_{\pi^* \in Q} \sum_{t=1}^T L\left(\pi^*, z^t\right) = \mathbf{Regret}^L(\mathcal{A}, Q, T),$$

*where $\pi^t = \mathcal{A}((\pi^j, z^j)_{j=1}^{t-1})$. When it is not clear from the context, we will use subscript to denote the setting – e.g. $\mathbf{Regret}^L_{\text{FAIR-BATCH}}$.*

We wish to develop an algorithm such that both the misclassfication and fairness loss regret is sublinear, which is often called no-regret. Note that because $\pi^* \in Q_\alpha$ is $\alpha$-fair on $\bar{x}^t$ for all $t \in [T]$, we have $\mathrm{Unfair}_\alpha(\pi^*, z^t) = 0$ for all $t \in [T]$. Hence, achieving $\mathbf{Regret}^{\mathrm{Unfair}_\alpha}_{\text{FAIR-BATCH}}(\mathcal{A}, Q, T) = o(T)$ is equivalent to ensuring that the total number of rounds with any $\alpha$-fairness violation is sublinear. Therefore, our goal is equivalent to developing an algorithm $\mathcal{A}$ so that for any $(z^t)_{t=1}^T$,

$$\mathbf{Regret}^{\mathrm{Err}}_{\text{FAIR-BATCH}}(\mathcal{A}, Q, T) = o(T) \quad \text{and} \quad \sum_{t=1}^T \mathrm{Unfair}_\alpha(\pi^t, z^t) = o(T).$$

To achieve the result above, we will reduce our setting to a setting with no fairness constraint, which we call *online batch classification* problem. Similar to the online fair batch classification setting, in each round $t$, the learner deploys a policy $\pi^t$, but the environment chooses only a batch of instances $(x_\tau^t, y_\tau^t)_{\tau=1}^k$. In online batch classification, we denote the strategy that the environment can take with $\mathcal{Z}_{\text{BATCH}} = \mathcal{X}^k \times \mathcal{Y}^k$. We compare the two settings in figure 1.

# 4 Achieving No Regret Simultaneously

Here, we define a round-based Lagrangian loss and show that the regret with respect to our Lagrangian loss also serves as the misclassification and the fairness complaint regret. Then, we show that using an auditor that can detect any fairness violation beyond certain threshold, we can still hope to achieve no-regret against an adaptive adversary.

Finally, we show how to achieve no regret with respect to the Lagrangian loss by reducing the problem to an online batch classification where there's no fairness constraint. We show that Follow-The-Perturbed-Leader style approach (CONTEXT-FTPL from Syrgkanis et al. (2016)) can achieve sublinear regret in the online batch classification setting, which allows us to achieve sublinear regret with respect to both misclassification and fairness loss in the online fair batch classification setting.

## 4.1 Lagrangian Formulation

Here we present a hybrid loss that we call *Lagrangian loss* that combines the misclassification loss and the magnitude of the fairness loss of round $t$.

**Definition 4.1** (Lagrangian Loss). *The $(C, \alpha)$-Lagrangian loss of $\pi$ is*

$$\mathcal{L}_{C,\alpha}\left(\pi, \left((\bar{x}^t, \bar{y}^t), \rho^t\right)\right) = \sum_{\tau=1}^{k} \ell\left(\pi\left(x_\tau^t\right), y_\tau^t\right) + \begin{cases} C\left(\pi(x_{\rho_1}^t) - \pi(x_{\rho_2}^t) - \alpha\right) & \rho^t = (\rho_1, \rho_2) \\ 0 & \rho^t = null \end{cases}$$

Given an auditor $\mathcal{J}_\alpha$ that can detect any $\alpha$-fairness violation, we can simulate the online fair batch classification setting with an auditor $\mathcal{J}_\alpha$ by setting the pair $\rho_{\mathcal{J}}^t = \mathcal{J}_\alpha(\bar{x}^t, \pi^t)$: subscript $\mathcal{J}$ is placed on this pair to distinguish from the pair chosen by the environment.[3]

**Definition 4.2** (Lagrangian Regret). *Algorithm $\mathcal{A}$'s $(C, \alpha, \mathcal{J}_{\alpha'})$-Lagrangian regret against $Q$ is*

$\mathbf{Regret}^{C,\alpha,\mathcal{J}_{\alpha'}}(\mathcal{A}, Q, T)$, *if for any* $(\bar{x}^t, \bar{y}^t)_{t=1}^T$, *we have*

$$\sum_{t=1}^{T} \mathcal{L}_{C,\alpha}(\pi^t, (\bar{x}^t, \bar{y}^t), \rho_{\mathcal{J}}^t) - \min_{\pi^* \in Q} \sum_{t=1}^{T} \mathcal{L}_{C,\alpha}(\pi^*, (\bar{x}^t, \bar{y}^t), \rho_{\mathcal{J}}^t) \leq \mathbf{Regret}^{C,\alpha,\mathcal{J}_{\alpha'}}(\mathcal{A}, Q, T),$$

*where* $\rho_{\mathcal{J}}^t = \mathcal{J}_{\alpha'}(\bar{x}^t, \pi^t)$.

**Remark 4.3.** *From here on, we assume the auditor has a given sensitivity denoted by $\alpha' = \alpha + \epsilon$, where $\epsilon$ is a parameter we will fix in order to define our desired benchmark $Q_\alpha$.*

Now, we show that the Lagrangian regret upper bounds the $\alpha$-fairness loss regret with some slack by setting $C$ to be appropriately big enough. Also, we show that $(C, \alpha, \mathcal{J}_{\alpha+\epsilon})$-Lagrangian regret serves as the misclassification loss regret, too. The proofs are given in Appendix A.1.

**Theorem 4.4.** *Fix some small constant $\epsilon > 0$ and $C \geq \frac{k+1}{\epsilon}$. For any sequence of environment's strategy $(z^t)_{t=1}^T \in \mathcal{Z}_{\text{FAIR-BATCH}}^T$, $\sum_{t=1}^{T} Unfair_{\alpha+\epsilon}(\pi^t, z^t) \leq \mathbf{Regret}^{C,\alpha,\mathcal{J}_{\alpha+\epsilon}}(\mathcal{A}, Q_\alpha, T)$.*

**Theorem 4.5.** *Fix some small constant $\epsilon > 0$. For any sequence of $(z^t)_{t=1}^T \in \mathcal{Z}_{\text{FAIR-BATCH}}^T$ and $\pi^* \in Q_\alpha$,*

$$\sum_{t=1}^{T} \sum_{\tau=1}^{k} \ell\left(\pi^t\left(x_\tau^t\right), y_\tau^t\right) - \sum_{t=1}^{T} \sum_{\tau=1}^{k} \ell(\pi^*(x_\tau^t), y_\tau^t) \leq \mathbf{Regret}^{C,\alpha,\mathcal{J}_{\alpha+\epsilon}}(\mathcal{A}, Q_\alpha, T),$$

*where $C \geq \frac{k+1}{\epsilon}$. In other words, $\mathbf{Regret}_{\text{FAIR-BATCH}}^{Err}(\mathcal{A}, Q_\alpha, T) \leq \mathbf{Regret}^{C,\alpha,\mathcal{J}_{\alpha+\epsilon}}(\mathcal{A}, Q_\alpha, T)$.*

## 4.2 Reduction to Online Batch Classification

In this subsection, we will first discuss a computationally inefficient way to achieve no regret with respect to the Lagrangian loss. Then, we will show an efficient reduction to online batch classification and discuss an example of an oracle-efficient algorithm $\mathcal{A}_{\text{BATCH}}$ that achieves no-regret.

It is well known that for linear loss, exponential weights with appropriately tuned learning rate $\gamma$ can achieve no regret (Freund and Schapire, 1997; Cesa-Bianchi et al., 1997; Arora et al., 2012). Note that our Lagrangian loss

$$\mathcal{L}_{C,\alpha}^t(\pi) = \mathcal{L}_{C,\alpha}(\pi, z^t) = \sum_{\tau=1}^{k} (1 - \pi(x_\tau^t)) \cdot y_\tau^t + \pi(x_\tau^t) \cdot (1 - y_\tau^t)$$

$$+ \begin{cases} C\left(\pi(x_{\rho_1}^t) - \pi(x_{\rho_2}^t) - \alpha\right) & \rho^t = (\rho_1, \rho_2) \\ 0 & \rho^t = null \end{cases}$$

is linear in $\pi$ for any $z^t$, and its range is $[0, C + k]$. Therefore, running exponential weights with learning rate $\gamma = \sqrt{\frac{\ln(|\mathcal{H}|)}{T}}$, we achieve the following regret with respect to the Lagrangian loss:

**Corollary 4.6.** *Running exponential weights with $\gamma = \sqrt{\frac{\ln(|\mathcal{H}|)}{T}}$ and $C \geq \frac{k+1}{\epsilon}$, we achieve*

$$\mathbf{Regret}_{\text{FAIR-BATCH}}^{Err}(\mathcal{A}, Q_\alpha, T) \leq (C+k)\sqrt{\ln(|\mathcal{H}|)T}, \quad \sum_{t=1}^T Unfair_{\alpha+\epsilon}(\pi^t, z^t) \leq (C+k)\sqrt{\ln(|\mathcal{H}|)T}.$$

Nevertheless, running exponential weights is not efficient as it needs to calculate the loss for each $h \in \mathcal{H}$ every round $t$. To design an oracle-efficient algorithm, we reduce the online batch fair classification problem to the online batch classification problem in an efficient manner and use any online batch algorithm $\mathcal{A}_{\text{BATCH}}((\pi^j, (\bar{x}'^j, \bar{y}'^j))_{j=1}^t)$ as a black box. At a high level, our reduction involves just carefully transforming our online fair batch classification history up to $t$, $(\pi^j, (\bar{x}^j, \bar{y}^j, \rho^j))_{j=1}^t \in (\Delta\mathcal{H} \times \mathcal{Z}_{\text{FAIR-BATCH}})^t$ into some fake online batch classification history $(\pi^j, (\bar{x}'^j, \bar{y}'^j))_{j=1}^t \in (\Delta\mathcal{H} \times \mathcal{Z}_{\text{BATCH}})^t$ and then feeding the artificially created history to $\mathcal{A}_{\text{BATCH}}$.

Without loss of generality, we assume that $C \geq \frac{k+1}{\epsilon}$ is an integer; if it's not, then take the ceiling. Now, we describe how the transformation of the history works. For each round $t$, whenever $\rho^t = (\rho_1^t, \rho_2^t)$, we add $C$ copies of each of $(x_{\rho_1^t}^t, 0)$ and $(x_{\rho_2^t}^t, 1)$ to the original pairs to form $\bar{x}'^t$ and $\bar{y}'^t$. Just to keep the batch size the same across each round, even if $\rho^t = null$, we add $C$ copies of each of $(v, 0)$ and $(v, 1)$ where $v$ is some arbitrary instance in $\mathcal{X}$. We describe this process in more detail in algorithm 3. This reduction essentially preserves the regret.

**Theorem 4.7.** *For any sequence of $(z^t)_{t=1}^T \in \mathcal{Z}_{\text{FAIR-BATCH}}^T$, $Q \subseteq \Delta\mathcal{H}$, and $\pi^* \in Q$,*

$$\sum_{t=1}^T \mathcal{L}_{C,\alpha}(\pi^t, z^t) - \sum_{t=1}^T \mathcal{L}_{C,\alpha}(\pi^*, z^t) \leq \mathbf{Regret}_{\text{BATCH}}^{Err}(\mathcal{A}, Q, T),$$

*where $\pi^t = \mathcal{A}_{\text{BATCH}}\left((\pi^j, \bar{x}'^j, \bar{y}'^j)_{j=1}^{t-1}\right)$. Therefore, $\mathbf{Regret}^{C,\alpha,\mathcal{J}_{\alpha+\epsilon}}(\mathcal{A}, Q_\alpha, T) \leq \mathbf{Regret}_{\text{BATCH}}^{Err}(\mathcal{A}, Q, T)$.*

One example of $\mathcal{A}_{\text{BATCH}}$ that achieves sublinear regret in online batch classification is CONTEXT-FTPL from Syrgkanis et al. (2016). We defer the details to Appendix A.3 and present the regret guarantee here. We only focus on their small separator set setting (i.e. there exists a small set of points which serves as a witness to distinguish any two different hypothesis), although their transductive setting (i.e. the contexts $\{x_t\}_{t=1}^T$ are known in advance) naturally follows as well.

**Theorem 4.8.** *If the separator set $S$ for $\mathcal{H}$ is of size $s$, then CONTEXT-FTPL achieves the following misclassification and fairness regret in the online fair batch classification setting.*

$$\mathbf{Regret}_{\text{FAIR-BATCH}}^{Err}(\mathcal{A}, Q_\alpha, T) \leq O\left(\left(\frac{sk}{\epsilon}\right)^{\frac{3}{4}} \sqrt{T\log(|\mathcal{H}|)}\right)$$

$$\sum_{t=1}^T Unfair_{\alpha+\epsilon}(\pi^t, z^t) \leq O\left(\left(\frac{sk}{\epsilon}\right)^{\frac{3}{4}} \sqrt{T\log(|\mathcal{H}|)}\right)$$

**Algorithm 3:** Reduction from Online Fair Batch Classification to Online Batch Classification

---

**Parameters:** inflation constant $C$, original round size $k$
**Initialize:** $k' = k + 2C$;
**for** $t = 1, \ldots, T$ **do**

> Learner deploys $\pi^t$;
> Environment chooses $(\bar{x}^t, \bar{y}^t)$ and the pair $\rho^t$;
> **if** $\rho^t = (\rho_1^t, \rho_2^t)$ **then**
>
> > **for** $i = 1, \ldots, C$ **do**
> >
> > > $x_{k+i}^t = x_{\rho_1^t}^t$    and    $y_{k+i}^t = 0$;
> > > $x_{k+C+i}^t = x_{\rho_2^t}^t$    and    $y_{k+C+i}^t = 1$;
> >
> > **end**
>
> **end**
> **else**
>
> > **for** $i = 1, \ldots, C$ **do**
> >
> > > $x_{k+i}^t = v$    and    $y_{k+i}^t = 0$;
> > > $x_{k+C+i}^t = v$    and    $y_{k+C+i}^t = 1$;
> >
> > **end**
>
> **end**
> $\bar{x}'^t = (x_\tau^t)_{\tau=1}^{k'}$    and    $\bar{y}'^t = (y_\tau^t)_{\tau=1}^{k'}$;
> $\pi^{t+1} = \mathcal{A}_{\text{BATCH}}\left( (\pi^j, \bar{x}'^j, \bar{y}'^j)_{j=1}^t \right)$;

**end**

---

# 5 Generalization

We observe that until this point, all of our results apply to the more general setting where individuals arrive in any adversarial fashion. In order to argue about generalization, in this section, we will assume the existence of an (unknown) data distribution from which individual arrivals are drawn: $\{\{(x_\tau^t, y_\tau^t)\}_{\tau=1}^k\}_{t=1}^T \sim_{i.i.d.} \mathcal{D}^{Tk}$.

Despite the data are drawn i.i.d., there are two technical challenges in establishing generalization guarantee: (1) the auditor's fairness feedback at each round is limited to a single fairness violation with regards to the policy deployed in that round, and (2) both the deployed policies and the auditor are adaptive over rounds. To overcome these challenges, we will draw a connection between the established regret guarantees in Section 4 and the learner's distributional accuracy and fairness guarantees. In particular, we will analyze the generalization bounds for the average policy over rounds.

**Definition 5.1** (Average Policy). *Let $\pi^t$ be the policy deployed by the algorithm at round $t$. The average policy $\pi^{avg}$ is defined by $\forall x : \pi^{avg}(x) = \frac{1}{T} \sum_{t=1}^T \pi^t(x)$.*

In order to be consistent with Section 4, we denote $\alpha' = \alpha + \epsilon$ in this section.

Here, we state the main results of this section:

**Theorem 5.2** (Accuracy Generalization). *With probabilty $1 - \delta$, the misclassification loss of $\pi^{avg}$ is upper bounded by*

$$\underset{(x,y)\sim\mathcal{D}}{\mathbb{E}}[\ell(\pi^{avg}(x), y)] \leq \inf_{\pi \in Q_\alpha} \underset{(x,y)\sim\mathcal{D}}{\mathbb{E}}[\ell(\pi(x), y)] + \frac{1}{kT}\mathbf{Regret}^{C,\alpha,\mathcal{J}_{\alpha+\epsilon}}(\mathcal{A}, Q_\alpha, T) + \sqrt{\frac{8\ln\left(\frac{4}{\delta}\right)}{T}}$$

**Theorem 5.3** (Fairness Generalization). *Assume that for all $t$, $\pi^t$ is $(\alpha, \beta^t)$-fair $(0 \leq \beta^t \leq 1)$. With probability $1 - \delta$, for any integer $q \leq T$, $\pi^{avg}$ is $(\alpha' + \frac{q}{T}, \beta^*)$-fair where*

$$\beta^* = \frac{1}{q}\left( \mathbf{Regret}^{C,\alpha,\mathcal{J}_{\alpha+\epsilon}}(\mathcal{A}, Q_\alpha, T) + \sqrt{2T\ln\left(\frac{2}{\delta}\right)} \right).$$

**Corollary 5.4.** *Using* CONTEXT-FTPL *from Syrgkanis et al. (2016) with a separator set of size $s$, with probability $1 - \delta$, the average policy $\pi^{avg}$ has the following guarantee:*

*1. Accuracy:*

$$\mathbb{E}_{(x,y)\sim\mathcal{D}}[\ell(\pi^{avg}(x),y)] \leq \inf_{\pi\in Q_\alpha} \mathbb{E}_{(x,y)\sim\mathcal{D}}[\ell(\pi(x),y)] + O\left(\frac{1}{k^{\frac{1}{4}}}\left(\frac{s}{\epsilon}\right)^{\frac{3}{4}}\sqrt{\frac{\ln(|\mathcal{H}|)+\ln\left(\frac{1}{\delta}\right)}{T}}\right).$$

*2. **Fairness:*** $\pi^{avg}$ *is* $(\alpha'+\lambda,\lambda)$*-fair where* $\lambda = O\left(\left(\frac{sk}{\epsilon}\right)^{\frac{3}{4}}\left(\frac{\ln(|\mathcal{H}|)+\ln\left(\frac{1}{\delta}\right)}{T}\right)^{\frac{1}{4}}\right).$

**Remark 5.5.** *Recall that the sensitivity of the auditor* $\alpha'$ *is fixed, and the learner chooses the parameter* $\epsilon \in (0,\alpha')$*, which in return determines* $\alpha = \alpha' - \epsilon$ *and the set of policy* $Q_\alpha$ *the learner is competing against. In the case where* $\alpha' = \Omega(1)$*, the learner can choose* $\epsilon$ *in the order of* $\Omega(1)$ *and guarantee that* $\pi^{avg}$ *is* $(\alpha'+\lambda,\lambda)$*-fair with* $\lambda = \tilde{O}(T^{-1/4})$*. In this regime, corollary 5.4 implies that policy* $\pi^{avg}$ *has a non-trivial accuracy guarantee and a fairness generalization bound that qualitatively matches the uniform convergence bound in Theorem 1.4 of Rothblum and Yona (2018).*

The accuracy generalization bound of Theorem 5.2 is attained by applying Azuma's inequality on the left hand side of the inequality in Theorem 4.5 and then leveraging the fact that our classification loss function is linear with respect to the base classifiers over which it is defined. The full proof is given in Appendix B.

As for the more challenging task of providing a fairness generalization guarantee (Theorem 5.3), we show how careful interpolation between $\alpha$ and $\beta$ may be be used to provide a meaningful bound. Here, we state the key lemma required for Theorem 5.3 and a brief description of the proof technique.

**Lemma 5.6.** *Assume that for all t,* $\pi^t$ *is* $(\alpha',\beta^t)$*-fair* $(0 \leq \beta^t \leq 1)$*. For any integer* $q \leq T$*,* $\pi^{avg}$ *is* $\left(\alpha'+\frac{q}{T},\frac{1}{q}\sum_{t=1}^{T}\beta^t\right)$*-fair.*

**High-Level Proof Idea for Lemma 5.6**   Setting $\alpha'' = \alpha' + \frac{q}{T}$ has the following implication: for any pair of individuals $(x,x')$, in order for $\pi^{avg}$ to have an $\alpha''$-fairness violation on $x,x'$, at least $q$ of the policies in $\{\pi^1,\ldots,\pi^T\}$ must have an $\alpha'$-fairness violation on $x,x'$. We will then say a subset $A \subseteq \mathcal{X} \times \mathcal{X}$ is $\alpha'$-covered by a policy $\pi$, if $\pi$ has an $\alpha'$-violation on every element in $A$. We will denote by $A_q^{\alpha'} \subseteq \mathcal{X} \times \mathcal{X}$ the subset of pairs of elements from $\mathcal{X}$ that are $\alpha'$-covered by at least $q$ policies in $\{\pi^1,\ldots,\pi^T\}$. Next, consider the probability space $\mathcal{D}|_\mathcal{X} \times \mathcal{D}|_\mathcal{X}$ over pairs of individuals. The lemma then follows from observing that for any $q \leq T$, $\Pr(A_q^{\alpha'}) \leq \frac{1}{q}\Pr(A_1^{\alpha'})$, as this will allow us to upper bound the probability of an $\alpha''$-fairness violation by the stated bound.

In Appendix B, we provide the full proof of Theorem 5.3, which features the covering argument presented in lemma 5.6, in addition to a concentration argument linking the probability of the algorithm deploying unfair policies throught its run to the regret guarantees proven in section 4. We also illustrate why an $\alpha,\beta$ interpolation is required in order to achieve a non-vacuous guarantee.

# 6   Conclusion

In this paper, we were able to answer an open question by Gillen et al. (2018), proving that online learning under an unknown individual fairness constraint is possible even without assuming a strong parametric form of the underlying similarity measure. We were further able to prove what we consider a very surprising generalization result, matching the state-of-the-art bounds for individual fairness given by Rothblum and Yona (2018), while eliminating or significantly relaxing all of their rather stringent assumptions. Contrary to previous work, which provided individual fairness generalization bounds utilizing standard uniform convergence arguments (Agarwal et al. (2018); Rothblum and Yona (2018)), we have presented a novel proof technique with the use of a composition covering argument (Lemma 5.6), we also believe is of separate interest.

# Broader Impact

As the authors of this work believe that bridging the gap between theoretical research in algorithmic fairness and practical use is of the essence, one of the main focuses of this work has been removing

the rather stringent assumptions made in previous research in individual fairness, and replacing these with more realistic ones (if any). As such, the contributions offered in the paper allow taking a step closer to incorporating the long sought-after notion of individual fairness into real life systems. The introduction of a fairness auditor gives a simple, elegant solution to the hurdle posed by the classic similarity metric assumption. The notion of individual fairness pursued in this work offers a strong guarantee on the individual's level (which is not given, for example, by the various more popular yet weaker notions of group fairness). We believe this combination between practicality of use and a strong fairness guarantee has the power to significantly impact our ability to ensure fairness and non-discrimination in machine learning based algorithms.

## Acknowledgments and Disclosure of Funding

We thank Sampath Kannan, Akshay Krishnamurthy, Katrina Ligett, and Aaron Roth for helpful conversations at an early stage of this work. Part of this work was done while YB, CJ, and ZSW were visiting the Simons Institute for the Theory of Computing. YB is supported in part by Israel Science Foundation (ISF) grant #1044/16, the United States Air Force and DARPA under contracts FA8750-16-C-0022 and FA8750-19-2-0222, and the Federmann Cyber Security Center in conjunction with the Israel national cyber directorate. CJ is supported in part by NSF grant AF-1763307. ZSW is supported in part by the NSF FAI Award #1939606, an Amazon Research Award, a Google Faculty Research Award, a J.P. Morgan Faculty Award, a Facebook Research Award, and a Mozilla Research Grant. Any opinions, findings and conclusions or recommendations expressed in this material are those of the author(s) and do not necessarily reflect the views of the United States Air Force and DARPA.

## Footnotes

[1]Rothblum and Yona (2018) show (Theorem 1.4 in their work) that if a policy $\pi$ is $\alpha$-fair on all pairs in a i.i.d. dataset of size $m$, then $\pi$ satisfies $\Pr_{x,x'}[|\pi(x) - \pi(x')| > \alpha + \epsilon] \leq \epsilon$, as long as $m \geq \tilde{\Omega}(1/\epsilon^4)$.

[2]We will overload the notation for this loss; regardless of what $\mathcal{Z}$ is, we'll assume $Err(\pi, z^t)$ is well-defined as long as $z^t$ includes $(\bar{x}^t, \bar{y}^t)$.

[3]Although we are simulating the adaptive environment's strategy $\rho^t$ with $\rho_{\mathcal{J}}^t$, note that the fairness loss with $\rho_{\mathcal{J}}^t$ will always be at least the fairness loss with $\rho^t$ because the auditor will always indicate if there's a fairness violation. This distinction between the pair chosen by the environment and the auditor is necessary just for technical reasons, as we need to ensure that the pair used to charge the Lagrangian loss incurs constant instantaneous regret in the rounds where there is actually some fairness violation, as the pair chosen by the environment can possibly have no fairness violation and hence negative instantaneous regret. This will be made more clear in the proof of Theorem 4.4.

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
