[Supplementary Material]

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

[4]This is the case where, for every pair of individuals $x, x'$ on which there exists a policy in $\{\pi^1, \ldots, \pi^T\}$ that has an $\alpha'$-fairness violation: (1) every policy $\pi \in \{\pi^1, \ldots, \pi^T\}$ that has an $\alpha$-fairness violation on $x, x'$ has a maximal violation (of value 1), (2) all non-violating policies (in $\{\pi^1, \ldots, \pi^T\}$) on $x, x'$ are arbitrarily close to the violation threshold $\alpha'$ on this pair, and (3) all of the directions of the policies' predictions' differences on $x, x'$ are correlated (no cancellations averaging over policies).

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

# A  Omitted Details from Section 4

## A.1  Omitted Details from Subsection 4.1

**Lemma A.1.** *Fix the sequence of the environment's strategies $(z^t)_{t=1}^T$ and hence $Q_\alpha$. For any $t \in [T]$, $\rho = (\rho_1, \rho_2)$, and $\pi^* \in Q_\alpha$,*

$$\mathcal{L}_{C,\alpha}(\pi^*, (\bar{x}^t, \bar{y}^t), \rho) \leq \sum_{\tau=1}^k \ell\left(\pi^*\left(x_\tau^t\right), y_\tau^t\right) + Cd(x_{\rho_1}^t, x_{\rho_2}^t).$$

*Proof.* We know that $\pi^* \in Q_\alpha$ must be $\alpha$-fair on $\bar{x}^t$ for any $t \in [T]$, meaning

$$\pi^*(x_{\rho_1}^t) - \pi^*(x_{\rho_2}^t) - d(x_{\rho_1}^t, x_{\rho_2}^t) - \alpha \leq 0$$

Thus, we have

$$\mathcal{L}_{C,\alpha}(\pi^*, (\bar{x}^t, \bar{y}^t), (\rho_1, \rho_2)) = \sum_{\tau=1}^k \ell\left(\pi^*\left(x_\tau^t\right), y_\tau^t\right) + C\left(\pi^*(x_{\rho_1}^t) - \pi(x_{\rho_2}^t) - \alpha\right)$$

$$\leq \sum_{\tau=1}^k \ell\left(\pi^*\left(x_\tau^t\right), y_\tau^t\right) + Cd(x_{\rho_1}^t, x_{\rho_2}^t)$$

$\square$

**Theorem 4.4.** *Fix some small constant $\epsilon > 0$ and $C \geq \frac{k+1}{\epsilon}$. For any sequence of environment's strategy $(z^t)_{t=1}^T \in \mathcal{Z}_{\text{FAIR-BATCH}}^T$, $\sum_{t=1}^T \text{Unfair}_{\alpha+\epsilon}(\pi^t, z^t) \leq \mathbf{Regret}^{C,\alpha,\mathcal{J}_{\alpha+\epsilon}}(\mathcal{A}, Q_\alpha, T)$.*

*Proof.* Fix $(z^t)_{t=1}^T = ((\bar{x}^t, \bar{y}^t) \times \rho^t)_{t=1}^T$ and any $\pi^* \in Q_\alpha$. Consider any round $t$ where there is some $(\alpha + \epsilon)$-fairness violation with respect to $\bar{x}^t$, meaning $\exists \tau, \tau' \in [k].v_{\alpha+\epsilon}(\pi^t, (x_\tau^t, x_{\tau'}^t)) > 0$. During those rounds, the auditor $\mathcal{J}_{\alpha+\epsilon}$ will report one of those violations $\rho_{\mathcal{J}}^t = (\rho_1^t, \rho_2^t)$. During these rounds with some $(\alpha + \epsilon)$-fairness violation, we show that the instantaneous regret with respect to the Lagrangian loss must be at least 1:

$$\mathcal{L}_{C,\alpha}(\pi^t, (\bar{x}^t, \bar{y}^t), \rho_{\mathcal{J}}^t) - \mathcal{L}_{C,\alpha}(\pi^*, (\bar{x}^t, \bar{y}^t), \rho_{\mathcal{J}}^t)$$

$$\geq \left(\sum_{\tau=1}^k \ell(\pi^t(x_\tau^t), y_\tau^t) + C\left(\pi(x_{\rho_1^t}^t) - \pi(x_{\rho_2^t}^t) - \alpha\right)\right) - \left(\sum_{\tau=1}^k \ell(\pi^*(x_\tau^t), y_\tau^t) + Cd(x_{\rho_1^t}^t, x_{\rho_2^t}^t)\right) \quad (1)$$

$$= \left(\sum_{\tau=1}^k \ell(\pi^t(x_\tau^t), y_\tau^t) - \sum_{\tau=1}^k \ell(\pi^*(x_\tau^t), y_\tau^t)\right) + C\left(\pi^t(x_{\rho_1^t}^t) - \pi^t(x_{\rho_2^t}^t) - d\left(x_{\rho_1^t}^t, x_{\rho_2^t}^t\right) - \alpha\right)$$

$$\geq -k + C\epsilon \quad (2)$$

$$\geq 1,$$

where (1) follows from Lemma A.1, and (2) from the fact that the pair $\rho_{\mathcal{J}}^t = (\rho_1^t, \rho_2^t)$ found by the auditor $\mathcal{J}_{\alpha+\epsilon}$ must have had $(\alpha + \epsilon)$-fairness violation, so the magnitude of $\alpha$-fairness violation must have been at least $\epsilon$.

Finally, we bound the number of these rounds by the Lagrangian regret.

$\mathbf{Regret}^{C,\alpha,\mathcal{J}_{\alpha+\epsilon}}\left(\mathcal{A},Q_\alpha,T\right)$

$$\geq \sum_{t=1}^{T} \mathcal{L}_{C,\alpha}(\pi^t,(\bar{x}^t,\bar{y}^t),\rho_\mathcal{J}^t) - \mathcal{L}_{C,\alpha}(\pi^*,(\bar{x}^t,\bar{y}^t),\rho_\mathcal{J}^t)$$

$$\geq \sum_{t=1}^{T} \left(\mathcal{L}_{C,\alpha}(\pi^t,(\bar{x}^t,\bar{y}^t),\rho_\mathcal{J}^t) - \mathcal{L}_{C,\alpha}(\pi^*,(\bar{x}^t,\bar{y}^t),\rho_\mathcal{J}^t)\right) \cdot \mathbb{1}\left(\exists\tau,\tau' \in [k].v_{\alpha+\epsilon}(\pi^t,(x_\tau^t,x_{\tau'}^t)) > 0\right)$$

$$\geq \sum_{t=1}^{T} \mathbb{1}\left(\exists\tau,\tau' \in [k].v_{\alpha+\epsilon}(\pi^t,(x_\tau^t,x_{\tau'}^t)) > 0\right)$$

$$\geq \sum_{t=1}^{T} \mathrm{Unfair}_{\alpha+\epsilon}(\pi^t,z^t)$$

Note that $z^t$ in the last inequality contains $\rho^t$ and not $\rho_\mathcal{J}^t$. $\qquad\square$

**Theorem 4.5.** *Fix some small constant $\epsilon > 0$. For any sequence of $(z^t)_{t=1}^{T} \in \mathcal{Z}_{\text{FAIR-BATCH}}^{T}$ and $\pi^* \in Q_\alpha$,*

$$\sum_{t=1}^{T}\sum_{\tau=1}^{k} \ell\left(\pi^t\left(x_\tau^t\right),y_\tau^t\right) - \sum_{t=1}^{T}\sum_{\tau=1}^{k}\ell(\pi^*(x_\tau^t),y_\tau^t) \leq \mathbf{Regret}^{C,\alpha,\mathcal{J}_{\alpha+\epsilon}}\left(\mathcal{A},Q_\alpha,T\right),$$

*where $C \geq \frac{k+1}{\epsilon}$. In other words, $\mathbf{Regret}_{\text{FAIR-BATCH}}^{Err}(\mathcal{A},Q_\alpha,T) \leq \mathbf{Regret}^{C,\alpha,\mathcal{J}_{\alpha+\epsilon}}\left(\mathcal{A},Q_\alpha,T\right)$.*

*Proof.* Fix $(z^t)_{t=1}^{T} = ((\bar{x}^t,\bar{y}^t) \times \rho^t)_{t=1}^{T}$ and $\pi^* \in Q_\alpha$.

It is sufficient to show that for any round $t \in [T]$,

$$\sum_{\tau=1}^{k} \ell\left(\pi^t\left(x_\tau^t\right),y_\tau^t\right) - \sum_{\tau=1}^{k}\ell(\pi^*(x_\tau^t),y_\tau^t) \leq \mathcal{L}_{C,\alpha}(\pi^t,(\bar{x}^t,\bar{y}^t),\rho_\mathcal{J}^t) - \mathcal{L}_{C,\alpha}(\pi^*,(\bar{x}^t,\bar{y}^t),\rho_\mathcal{J}^t).$$

If $\rho_\mathcal{J}^t = null$, then the equality holds, so assume $\rho_\mathcal{J}^t = (\rho_1,\rho_2)$. In that case, because the pair was found by $\mathcal{J}_{\alpha+\epsilon}$, we have

$$\pi^t(x_{\rho_1}^t) - \pi^t(x_{\rho_2}^t) - d(x_{\rho_1}^t,x_{\rho_2}^t) - \alpha > \pi^t(x_{\rho_1}^t) - \pi^t(x_{\rho_2}^t) - d(x_{\rho_1}^t,x_{\rho_2}^t) - \alpha - \epsilon > 0$$

Then, we know

$$\sum_{\tau=1}^{k} \ell\left(\pi^t\left(x_\tau^t\right),y_\tau^t\right) - \sum_{\tau=1}^{k}\ell(\pi^*(x_\tau^t),y_\tau^t)$$

$$\leq \sum_{\tau=1}^{k} \ell\left(\pi^t\left(x_\tau^t\right),y_\tau^t\right) + C\left(\pi^t(x_{\rho_1}^t) - \pi^t(x_{\rho_2}^t) - d(x_{\rho_1}^t,x_{\rho_2}^t) - \alpha\right) - \sum_{\tau=1}^{k}\ell(\pi^*(x_\tau^t),y_\tau^t)$$

$$\leq \mathcal{L}_{C,\alpha}(\pi^t,(\bar{x}^t,\bar{y}^t),\rho_\mathcal{J}^t) - \mathcal{L}_{C,\alpha}(\pi^*,(\bar{x}^t,\bar{y}^t),\rho_\mathcal{J}^t) \qquad (3)$$

where (3) follows from Lemma A.1. $\qquad\square$

## A.2 Omitted Details from Subsection 4.2

**Theorem A.2** (Arora et al. (2012)). *For a linear loss function $L^t(\pi) = L(\pi,z^t) \in [0,M]$, exponential weights with learning rate $\gamma$ has the following guarantee: for any sequence of $(L^t)_{t=1}^{T}$ and for any other $\pi \in \Delta\mathcal{H}$,*

$$\sum_{t=1}^{T} L^t(\pi^t) \leq \sum_{t=1}^{T} L^t(\pi) + M \cdot \left(\frac{\ln(|\mathcal{H}|)}{\gamma} + \gamma T\right),$$

*where $\pi^0$ is a uniform distribution over $\mathcal{H}$ and $\pi_h^{t+1} \propto \exp(-\gamma \cdot L^t(h)) \cdot \pi_h^t$ for each $h \in \mathcal{H}$.*

**Lemma A.3.** *For any sequence of $(\pi^t)_{t=1}^T$, $(z^t)_{t=1}^T \in \mathcal{Z}_{\text{FAIR-BATCH}}^T$, and $\pi^* \in \Delta\mathcal{H}$,*

$$\sum_{t=1}^T \mathcal{L}_{C,\alpha}(\pi^t, z^t) - \sum_{t=1}^T \mathcal{L}_{C,\alpha}(\pi^*, z^t) = \sum_{t=1}^T \sum_{\tau=1}^{k'} \ell(\pi^t(x_\tau'^t), y_\tau'^t) - \sum_{t=1}^T \sum_{\tau=1}^{k'} \ell(\pi^*(x_\tau'^t), y_\tau'^t)$$

*Proof.* It is sufficient to show that in each round $t$,

$$\mathcal{L}_{C,\alpha}(\pi^t, z^t) - \mathcal{L}_{C,\alpha}(\pi^*, z^t) = \sum_{\tau=1}^{k'} \ell(\pi(x_\tau^t), y_\tau^t) - \sum_{\tau=1}^{k'} \ell(\pi(x_\tau^*), y_\tau^t)$$

First, assume $\rho^t = (\rho_1^t, \rho_2^t)$.

$$\mathcal{L}_{C,\alpha}(\pi^t, z^t) - \mathcal{L}_{C,\alpha}(\pi^*, z^t)$$

$$= \left( \sum_{\tau=1}^k \ell(\pi^t(x_\tau^t), y_\tau^t) + C(\pi^t(x_{\rho_1^t}^t) - \pi^t(x_{\rho_2^t}^t) - \alpha) \right) - \left( \sum_{\tau=1}^k \ell(\pi^*(x_\tau^t), y_\tau^t) + C(\pi^*(x_{\rho_1^t}^t) - \pi^*(x_{\rho_2^t}^t) - \alpha) \right)$$

$$= \left( \sum_{\tau=1}^k \ell(\pi^t(x_\tau^t), y_\tau^t) + \left( \sum_{\tau=1}^C \ell(\pi^t(x_{\rho_1^t}^t), 0) + \sum_{\tau=1}^C \ell(\pi^t(x_{\rho_2^t}^t), 1) - C \right) \right)$$

$$- \left( \sum_{\tau=1}^k \ell(\pi^*(x_\tau^t), y_\tau^t) + \left( \sum_{\tau=1}^C \ell(\pi^*(x_{\rho_1^t}^t), 0) + \sum_{\tau=1}^C \ell(\pi^*(x_{\rho_2^t}^t), 1) - C \right) \right)$$

$$= \sum_{\tau=1}^{k'} \ell(\pi^t(x_\tau'^t), y_\tau'^t) - \sum_{\tau=1}^{k'} \ell(\pi^*(x_\tau'), y_\tau'^t),$$

The second equality follows from the fact that for any $\pi$ and $x$,

$$\ell(\pi(x), 0) = \pi(x) \quad \text{and} \quad \ell(\pi(x), 1) = 1 - \pi(x).$$

If $\rho^t = null$, then the same argument applies as above; the only difference is that all the $\pi^t(v)$ will cancel with each other because the number of copies with label 0 is exactly the same as that of label 1. $\square$

**Theorem 4.7.** *For any sequence of $(z^t)_{t=1}^T \in \mathcal{Z}_{\text{FAIR-BATCH}}^T$, $Q \subseteq \Delta\mathcal{H}$, and $\pi^* \in Q$,*

$$\sum_{t=1}^T \mathcal{L}_{C,\alpha}(\pi^t, z^t) - \sum_{t=1}^T \mathcal{L}_{C,\alpha}(\pi^*, z^t) \leq \mathbf{Regret}_{\text{BATCH}}^{Err}(\mathcal{A}, Q, T),$$

*where $\pi^t = \mathcal{A}_{\text{BATCH}}\left((\pi^j, \bar{x}'^j, \bar{y}'^j)_{j=1}^{t-1}\right)$.  Therefore,  $\mathbf{Regret}^{C,\alpha,\mathcal{J}_{\alpha+\epsilon}}(\mathcal{A}, Q_\alpha, T) \leq \mathbf{Regret}_{\text{BATCH}}^{Err}(\mathcal{A}, Q, T)$.*

*Proof.* Fix the sequence $(z^t)_{t=1}^T \in \mathcal{Z}_{\text{FAIR-BATCH}}^T$. Due to Lemma A.3 and the definition of regret, we have that for any $Q \subseteq \Delta\mathcal{H}$ and $\pi^* \in Q$,

$$\sum_{t=1}^T \mathcal{L}_{C,\alpha}(\pi^t, z^t) - \sum_{t=1}^T \mathcal{L}_{C,\alpha}(\pi^*, z^t) = \sum_{t=1}^T \sum_{\tau=1}^{k'} \ell(\pi^t(x_\tau'^t), y_\tau'^t) - \sum_{t=1}^T \sum_{\tau=1}^{k'} \ell(\pi^*(x_\tau'^t), y_\tau'^t) \leq \mathbf{Regret}_{\text{BATCH}}^{Err}(\mathcal{A}, Q, T),$$

meaning $\mathbf{Regret}_{\text{BATCH}}^{Err}(\mathcal{A}, Q, T)$ serves as an upper bound on $\mathbf{Regret}^{C,\alpha,\mathcal{J}_{\alpha+\epsilon}}(\mathcal{A}, Q, T)$.

Also, since $Q_\alpha \subseteq Q$, we have

$$\mathbf{Regret}^{C,\alpha,\mathcal{J}_{\alpha+\epsilon}}(\mathcal{A}, Q_\alpha, T) \leq \mathbf{Regret}^{C,\alpha,\mathcal{J}_{\alpha+\epsilon}}(\mathcal{A}, Q, T) \leq \mathbf{Regret}_{\text{BATCH}}^{Err}(\mathcal{A}, Q, T).$$

$\square$

### A.3 Using CONTEXT-FTPL from Syrgkanis et al. (2016)

Syrgkanis et al. (2016) considers an adversarial contextual learning setting where in each round $t$, the learner randomly deploys some policy $\psi^t$, and the environment chooses $(\xi^t, \bar{y}^t) \in \Xi \times \{0,1\}^k$, where $k$ indicates the number of possible actions that can be taken for the instance $\xi^t$. The only knowledge at round $t$ not available to the environment is the randomness over how the learner chooses $\psi^t$. And there's some linear loss $L(\psi^t(\xi), \bar{y}^t)$.

They show that in the small separator setting, they can achieve sublinear regret given that they can compute a separator set prior to learning. We first define the definition of a separator set and then state their regret guarantee.

**Definition A.4.** *A set $S = (\xi_1, \ldots, \xi_n)$ is called a* separator set *for $\Psi : \Xi \to \{0,1\}^k$ if for any different $\psi$ and $\psi'$ in $\Psi$, there exists $\xi \in S$ such that $\psi(\xi) \neq \psi'(\xi)$.*

**Theorem A.5** (Syrgkanis et al. (2016)). *For any sequence of $(\bar{x}^t, \bar{y}^t)_{t=1}^T$, CONTEXT-FTPL$(S, \omega)$ initialized with a separator set $S$ and parameter $\omega$ achieves the following regret:*

$$\mathbb{E}\left[\sum_{t=1}^T L(\psi^t(\xi^t), \bar{y}^t) - \sum_{t=1}^T L(\psi^t(\xi^t), \bar{y}^t)\right] \leq 4\omega k n \sum_{t=1}^T \mathbb{E}[\|L(\cdot, \bar{y}^t)\|_*^2] + \frac{10}{\omega}\sqrt{nk}\log(|\mathcal{H}|),$$

*where $n = |S|$, $\|L(\cdot, \bar{y}^t)\|_* = \max_{\hat{y} \in \{0,1\}^k} L(\hat{y}, \bar{y}^t)$, and the expectation is over the algorithm* CONTEXT-FTPL.

Our online batch classification setting can be easily reduced to their setting by simply considering the batch of instances $\bar{x}^t$ as one single instance, meaning we set $\Xi = \mathcal{X}^k$. And we view each policy as $\psi_h(x^t) = (h(x_1^t), \ldots, h(x_k^t))$. In other words, we can define the policy class induced by $\mathcal{H}$ as

$$\Psi_{\mathcal{H}} = \left\{\forall h \in \mathcal{H} : (x_\tau)_{\tau=1}^k \mapsto (h(x_\tau))_{\tau=1}^k\right\}.$$

Therefore, the random policy $\psi_\pi \in \Delta\Psi_{\mathcal{H}}$ can be decomposed into convex combination of $\psi_h$'s. Finally, we construct our linear loss function as $L\left((\pi(x_\tau))_{\tau=1}^k, \bar{y}\right) = \langle(\pi(x_\tau))_{\tau=1}^k, 1 - 2\bar{y}\rangle$. Note that $\mathbb{E}_{h \sim \pi}[L(\psi_h(\xi^t), \bar{y})] = L(\psi_\pi(\xi^t), \bar{y})$ by linearity of expectation.

Furthermore, we can turn any separator set $S \subseteq \mathcal{X}$ for $\mathcal{H}$ into an equal size separator set $S' \subseteq \Xi$ for $\Psi$. In fact, the construction is as follows:

$$S' = \{\forall x \in S : \xi_x = (x, v, \ldots, v)\},$$

where $v$ is some arbitrary instance in $\mathcal{X}$.

**Lemma A.6.** *If $S$ is the separator set for $\mathcal{H}$, then $S'$ is a separator set for $\Psi$.*

*Proof.* Fix any $h$ and $h'$. Note that by definition of $S$, there exists $x \in S$ such that $h(x) \neq h'(x)$. As a result, $\psi_h(\xi_x) \neq \psi_{h'}(\xi_x)$ as $(h(x), q, \ldots, q) \neq (h(x'), q, \ldots, q)$. $\square$

Note that the reduction preserves the loss difference any $\pi$ and $\pi'$.

**Lemma A.7.**

$$L(\psi_\pi(\bar{x}), \bar{y}) - L(\psi_{\pi'}(\bar{x}), \bar{y}) = \sum_{\tau=1}^k \ell(\pi(x_\tau^t), y_\tau^t) - \sum_{\tau=1}^k \ell(\pi'(x_\tau^t), y_\tau^t)$$

*Proof.*

$$L(\psi_\pi(\bar{x}), \bar{y}) - L(\psi_{\pi'}(\bar{x}), \bar{y})$$
$$= \langle(\pi(x_\tau))_{\tau=1}^k, 1 - 2\bar{y}\rangle - \langle(\pi'(x_\tau))_{\tau=1}^k, 1 - 2\bar{y}\rangle$$
$$= \left(\sum_{\tau=1}^k (1 - \pi(x_\tau)) \cdot y_\tau + \pi(x_\tau) \cdot (1 - y_\tau)\right) - \left(\sum_{\tau=1}^k (1 - \pi'(x_\tau)) \cdot y_\tau + \pi(x_\tau) \cdot (1 - y_\tau)\right)$$
$$= \sum_{\tau=1}^k \ell(\pi(x_\tau^t), y_\tau^t) - \sum_{\tau=1}^k \ell(\pi'(x_\tau^t), y_\tau^t)$$

$\square$

**Theorem A.8.** *If the separator set $S$ for $\mathcal{H}$ is of size $s$,* CONTEXT-FTPL *achieves the following regret in the online batch classification setting:*

$$\sum_{t=1}^{T}\sum_{\tau=1}^{k}\ell(\pi^t(x_\tau^t), y_\tau^t) - \sum_{t=1}^{T}\sum_{\tau=1}^{k}\ell(\pi^*(x_\tau^t), y_\tau^t)$$

$$\leq \mathbf{Regret}_{\text{BATCH}}^{Err}(\mathcal{A}, Q, T) = O\left((ks)^{\frac{3}{4}}\sqrt{T\log(|\mathcal{H}|)}\right)$$

*Proof.* Since we can construct a separator set $S'$ for $\Psi$ (Lemma A.7), we can run CONTEXT-FTPL$(S', \omega)$ with $\omega$ appropriately set to achieve the above regret (Theorem A.5). $\square$

Using CONTEXT-FTPL as a black box for $\mathcal{A}_{\text{BATCH}}$ to solve the online fair batch classification, we get the following regret for the misclassification and fairness loss regret.

**Remark A.9.** *Note that in each round $t$,* CONTEXT-FTPL *simply returns a base policy $\psi_h$ such that single $h$ will be used to classify everyone in the batch $\bar{x}^t$. However, in some cases, it might be more desirable to deploy a randomized policy $\pi$ such that $h$ is sampled from $\pi$ for each individual $x_\tau^t$ in the batch. In such cases, one can either run* CONTEXT-FTPL *for each $x_\tau^t$ or run* CONTEXT-FTPL *multiple times to form a uniform mixture of $h$'s that approximates $\pi$.*

**Theorem 4.8.** *If the separator set $S$ for $\mathcal{H}$ is of size $s$, then* CONTEXT-FTPL *achieves the following misclassification and fairness regret in the online fair batch classification setting.*

$$\mathbf{Regret}_{\text{FAIR-BATCH}}^{Err}(\mathcal{A}, Q_\alpha, T) \leq O\left(\left(\frac{sk}{\epsilon}\right)^{\frac{3}{4}}\sqrt{T\log(|\mathcal{H}|)}\right)$$

$$\sum_{t=1}^{T}\mathit{Unfair}_{\alpha+\epsilon}(\pi^t, z^t) \leq O\left(\left(\frac{sk}{\epsilon}\right)^{\frac{3}{4}}\sqrt{T\log(|\mathcal{H}|)}\right)$$

*Proof.* The proof follows from Theorem A.8 along with the fact that the batch size is $k + 2C$, where $C \geq \frac{k+1}{\epsilon}$. $\square$

# B   Omitted Details from Section 5

## B.1   Accuracy Generalization

Here we first provide all the missing details for the accuracy generalization.

**Theorem 5.2** (Accuracy Generalization). *With probabilty $1 - \delta$, the misclassification loss of $\pi^{avg}$ is upper bounded by*

$$\mathbb{E}_{(x,y)\sim\mathcal{D}}[\ell(\pi^{avg}(x), y)] \leq \inf_{\pi\in Q_\alpha}\mathbb{E}_{(x,y)\sim\mathcal{D}}[\ell(\pi(x), y)] + \frac{1}{kT}\mathbf{Regret}^{C,\alpha,\mathcal{J}_{\alpha+\epsilon}}(\mathcal{A}, Q_\alpha, T) + \sqrt{\frac{8\ln\left(\frac{4}{\delta}\right)}{T}}$$

*Proof.* By Theorem 4.5, for any sequence of $(z^t)_{t=1}^{T}$ and $\pi^* \in Q_\alpha$,

$$\sum_{t=1}^{T}\sum_{\tau=1}^{k}\ell\left(\pi^t\left(x_\tau^t\right), y_\tau^t\right) - \ell(\pi^*(x_\tau^t), y_\tau^t) \leq \mathbf{Regret}^{C,\alpha,\mathcal{J}_{\alpha+\epsilon}}(\mathcal{A}, Q_\alpha, T)$$

We select $\pi^* \in \operatorname*{argmin}_{\pi\in Q^\alpha}\mathbb{E}_{(x,y)\sim\mathcal{D}}[\ell(\pi(x), y)]$. Also, fix the sequence of $(z^t)_{t=1}^{T}$ and the random coin flips of the algorithm $\mathcal{A}$. Therefore, $(\pi^t)_{t=1}^{T}$ must be fixed as well.

**Lemma B.1.**

$$\Pr_{\substack{(\bar{x}^t,\bar{y}^t)_{t=1}^{T} \\ \pi^t=\mathcal{A}(\cdots)}}\left[\left|\sum_{t=1}^{T}\sum_{\tau=1}^{k}\ell\left(\pi^t\left(x_\tau^t\right), y_\tau^t\right) - \mathbb{E}_{(\bar{x}'^t,\bar{y}'^t)_{t=1}^{T}}\left[\sum_{t=1}^{T}\sum_{\tau=1}^{k}\ell\left(\pi^t\left(x_\tau'^t\right), y_\tau'^t\right)\right]\right| \geq \gamma\right] \leq 2\exp\left(\frac{-\gamma^2}{2k^2T}\right)$$

*Proof.* Define $(A^t)_{t=1}^T$ as

$$A^t = \sum_{j=1}^t \sum_{\tau=1}^k \ell\left(\pi^j\left(x_\tau^j\right), y_\tau^j\right) - \underset{(\bar{x}'^j, \bar{y}'^j)_{j=1}^t}{\mathbb{E}} \left[\sum_{j=1}^t \sum_{\tau=1}^k \ell\left(\pi^j\left(x_\tau'^j\right), y_\tau'^j\right)\right].$$

Note that $(A^t)_{t=1}^T$ is a martingale as $\mathbb{E}[A^t | A^1, \ldots, A^{t-1}] = A^{t-1}$ because $\pi^t$ is determined deterministically in terms of the previous history.

Note that $|A^t - A^{t-1}| \le k$. Therefore, applying Azuma's inequality yields

$$\Pr\left[|A^T - A^1| \ge \gamma\right] \le 2\exp\left(\frac{-\gamma^2}{2k^2 T}\right).$$

$\square$

Applying the Chernoff bound, we get a similar concentration bound on $\pi^*$:

$$\underset{(\bar{x}^t, \bar{y}^t)_{t=1}^T}{\Pr}\left[\left|\sum_{t=1}^T \sum_{\tau=1}^k \ell\left(\pi^*\left(x_\tau^t\right), y_\tau^t\right) - \underset{(\bar{x}'^t, \bar{y}'^t)_{t=1}^T}{\mathbb{E}}\left[\sum_{t=1}^T \sum_{\tau=1}^k \ell\left(\pi^*\left(x_\tau'^t\right), y_\tau'^t\right)\right]\right| \ge \gamma\right] \le 2\exp\left(\frac{-\gamma^2}{2k^2 T}\right).$$

Next, using triangle inequality, with probability $1 - \delta$, it holds that

$$\underset{(\bar{x}^t, \bar{y}^t)_{t=1}^T}{\mathbb{E}}\left[\sum_{t=2}^T \sum_{\tau=1}^k \ell\left(\pi^t\left(x_\tau^t\right), y_\tau^t\right)\right] - \underset{(\bar{x}^t, \bar{y}^t)_{t=1}^T}{\mathbb{E}}\left[\sum_{t=1}^T \sum_{\tau=1}^k \ell(\pi^*(x_\tau^t), y_\tau^t)\right]$$

$$\le \mathbf{Regret}^{C,\alpha,\mathcal{J}_{\alpha+\epsilon}}(\mathcal{A}, Q_\alpha, T) + 2\sqrt{\ln\left(\frac{4}{\delta}\right) 2k^2 T}$$

Equivalently, with probability $1 - \delta$,

$$\underset{(\bar{x}^t, \bar{y}^t)_{t=1}^T}{\mathbb{E}}\left[\frac{1}{kT}\sum_{t=1}^T \sum_{\tau=1}^k \ell\left(\pi^t\left(x_\tau^t\right), y_\tau^t\right)\right] \le \min_{\pi \in Q_\alpha} \underset{(x,y)\sim\mathcal{D}}{\mathbb{E}}[\ell(\pi(x), y)]$$

$$+ \frac{1}{kT}\mathbf{Regret}^{C,\alpha,\mathcal{J}_{\alpha+\epsilon}}(\mathcal{A}, Q_\alpha, T) + 2\sqrt{\frac{2\ln\left(\frac{4}{\delta}\right)}{T}}$$

For the left hand side on the inequality, observe that the following holds:

$$\underset{(x,y)\sim\mathcal{D}}{\mathbb{E}}[\ell(\pi^{avg}(x), y)] = \frac{1}{kT}\sum_{\tau=1}^k \sum_{i=1}^T \underset{(x,y)\sim\mathcal{D}}{\mathbb{E}}\left[\underset{\pi\sim\mathbb{U}\{\pi^1,\ldots,\pi^T\}}{\mathbb{E}}[\ell(\pi(x), y)]\right] \qquad (4)$$

$$= \frac{1}{kT}\sum_{\tau=1}^k \sum_{i=1}^T \underset{(x,y)\sim\mathcal{D}}{\mathbb{E}}\left[\frac{1}{T}\sum_{t=1}^T \ell(\pi^t(x), y)\right]$$

$$= \frac{1}{kT}\sum_{\tau=1}^k \sum_{t=1}^T \underset{(x,y)\sim\mathcal{D}}{\mathbb{E}}\left[\frac{1}{T}\sum_{i=1}^T \ell(\pi^t(x), y)\right]$$

$$= \frac{1}{kT}\sum_{\tau=1}^k \sum_{t=1}^T \underset{(x,y)\sim\mathcal{D}}{\mathbb{E}}\left[\ell(\pi^t(x), y)\right]$$

$$= \frac{1}{kT} \underset{(\bar{x}^t, \bar{y}^t)_{t=1}^T \sim_{i.i.d.} \mathcal{D}^{kT}}{\mathbb{E}}\left[\sum_{t=1}^T \sum_{\tau=1}^k \ell(\pi^t(x_\tau^t), y_\tau^t)\right]$$

Transition 4 stems from the fact that our misclassification loss $\ell$ is linear with respect to the base classifiers in $\mathcal{H}$. Hence, taking the uniform distribution over $\pi^1, \ldots, \pi^T$ gives

$$\ell(\pi^{avg}(x), y) = \underset{\pi\sim\mathbb{U}\{\pi^1,\ldots,\pi^T\}}{\mathbb{E}}[\ell(\pi(x), y)].$$

$\square$

## B.2 Fairness Generalization

A more challenging task for generalization is to argue about the fairness generalization guarantee of the average policy (Theorem 5.3). To provide some intuition for why this is the case, let us first attempt to upper bound the probability of running into an $\alpha'$-fairness violation by the average policy on a randomly selected pair of individuals:

**Observation B.2.** *Suppose for all $t$, $\pi^t$ is $(\alpha', \beta^t)$-fair. Then, $\pi^{avg}$ is $\left(\alpha', \sum_{t=1}^{T} \beta^t\right)$-fair.*

This bound is very dissatisfying, as the statement is vacuous when $\sum_{t=1}^{T} \beta^t \geq 1$. The reason for such a weak guarantee is that by aiming to upper bound the unfairness probability for the original fairness violation threshold $\alpha'$, we are subject to worst-case compositional guarantees[4] in the sense that the average policy may result to have an $\alpha'$-fairness violation on any fraction of the distribution (over pairs) where one or more of the deployed policies induces an $\alpha'$-fairness violation. This bound is tight, as we will see next.

**Tightness of Bound in Observation B.2**
Consider the following example: $\mathcal{X} = \{x_1, x_2, x_2\}$, $\mathcal{D}|_{\mathcal{X}} = \mathbb{U}\{\mathcal{X}\}$ (i.e. uniform distribution over $\mathcal{X}$), $\alpha' = 0.1$. $\mathcal{H} = \{h_1, h_2\}$ given by:

$$h_1(x_1) = 1, \; h_1(x_2) = 0, \; h_1(x_3) = 0$$
$$h_2(x_1) = 1, \; h_2(x_2) = 1, \; h_2(x_3) = 0$$

Also, assume the dissimilarity measure by the auditor is:

$$d(x_1, x_2) = 0, \; d(x_2, x_3) = 0, \; d(x_3, x_1) = 1.$$

Assume the algorithm deploys policies in the following manner: $\pi^t = \begin{cases} h_1 & t \text{ is odd} \\ h_2 & t \text{ is even} \end{cases}$

Both $h_1, h_2$ are exactly $(\alpha', \frac{1}{8}) - fair$. If $T$ is even, $\pi^{avg}$ is exactly $(\alpha', \frac{1}{4})$-fair.

**Interpolating $\alpha$ and $\beta$**  To circumvent the above setback, our strategy will be to relax the target violation threshold of the desired fairness guarantee of the average policy to $\alpha'' > \alpha'$. How big should we set $\alpha''$? A good intuition may arrive from considering the following thought experiment: assume worst-case compositional guarantees, and then, select a pair of individuals $x, x'$ on which the average policy has an $\alpha''$-fairness violation. We aim to lower bound the number of policies from $\{\pi^1, \ldots, \pi^T\}$ that have an $\alpha'$-fairness violation on this pair. As we will see, setting $\alpha''$ to be sufficiently larger will force the number of these policies required to produce an $\alpha''$-fairness violation of the average policy on $x, x'$ to be high, resulting in the following improved bound:

**Lemma 5.6.** *Assume that for all $t$, $\pi^t$ is $(\alpha', \beta^t)$-fair $(0 \leq \beta^t \leq 1)$. For any integer $q \leq T$, $\pi^{avg}$ is $\left(\alpha' + \frac{q}{T}, \frac{1}{q} \sum_{t=1}^{T} \beta^t\right)$-fair.*

*Proof.* Fix $\{\pi^t\}_{t=1}^T$, and assume that $\forall t : \pi^t$ is $(\alpha', \beta^t)$-fair. If we set $q \le T$, then we know that

$$\Pr_{x,x'}\left[|\pi^{avg}(x) - \pi^{avg}(x')| - d(x,x') > \alpha' + \frac{q}{T}\right]$$

$$= \Pr_{x,x'}\left[\left|\frac{1}{T}\sum_{t=1}^T [\pi^t(x) - \pi^t(x')]\right| - d(x,x') > \alpha' + \frac{q}{T}\right]$$

$$\le \Pr_{x,x'}\left[\left[\frac{1}{T}\sum_{t=1}^T |\pi^t(x) - \pi^t(x')|\right] - d(x,x') > \alpha' + \frac{q}{T}\right]$$

$$\le \Pr_{x,x'}\left[\exists\{i_1,\ldots,i_q\} \subseteq [T], \forall j, |\pi^{i_j}(x) - \pi^{i_j}(x')| - d(x,x') > \alpha'\right] \qquad (5)$$

$$\le \frac{1}{q}\sum_{t=1}^T \Pr_{x,x'}\left[|\pi^t(x) - \pi^t(x')| - d(x,x') > \alpha'\right] \qquad (6)$$

$$\le \frac{1}{q}\sum_{t=1}^T \beta^t$$

Transition 5 is given by the following observation: fix any $x, x'$ and assume

$$\left|\{\pi^t : t \in [T], |\pi^t(x) - \pi^t(x')| - d(x,x') > \alpha'\}\right| \le q$$

Then, we have

$$|\pi^{avg}(x) - \pi^{avg}(x')| - d(x,x') \le \frac{q + (T-q)\alpha'}{T} = \alpha' + \frac{q}{T} - \frac{\alpha'q}{T} < \alpha' + \frac{q}{T}.$$

Transition 6 stems from the following argument: for any $x, x'$, denote by

$$V_{x,x'}^{\alpha'} := \{\pi^t : t \in [T], |\pi^t(x) - \pi^t(x')| - d(x,x') > \alpha'\}$$

the subset of deployed policies that have an $\alpha'$-fairness violation on $x, x'$. We know that

$$\frac{1}{q}\sum_{t=1}^T \Pr_{x,x'}\left[|\pi^t(x) - \pi^t(x')| - d(x,x') > \alpha'\right]$$

$$= \frac{1}{q}\sum_{t=1}^T \int_x \int_{x'} \mathbb{P}(x,x') \cdot \mathbb{1}[|\pi^t(x) - \pi^t(x')| - d(x,x') > \alpha']dx'dx$$

$$= \frac{1}{q}\int_x \int_{x'} \sum_{t=1}^T \mathbb{P}(x,x') \cdot \mathbb{1}[|\pi^t(x) - \pi^t(x')| - d(x,x') > \alpha']dx'dx$$

$$= \frac{1}{q}\int_x \int_{x'} \sum_{\pi^t \in V_{x,x'}^{\alpha'} : \left|V_{x,x'}^{\alpha'}\right| \ge q} \mathbb{P}(x,x') + \sum_{\pi^t \in V_{x,x'}^{\alpha'} : \left|V_{x,x'}^{\alpha'}\right| < q} \mathbb{P}(x,x')dx'dx$$

$$\ge \int_x \int_{x'} \mathbb{P}(x,x') \cdot \mathbb{1}\left[|V_{x,x'}^{\alpha'}| \ge q\right] dx'dx$$

$$= \Pr_{x,x'}\left[\exists\{i_1,\ldots,i_q\} \subseteq [T], \forall j, |\pi^{i_j}(x) - \pi^{i_j}(x')| - d(x,x') > \alpha'\right],$$

where $\mathbb{P}(x,x')$ denotes the probability measure of $x, x'$ defined by $\mathcal{D}|_{\mathcal{X}} \times \mathcal{D}|_{\mathcal{X}}$. This concludes the proof. $\qquad\square$

**Lemma B.3.** *With probability $1 - \delta$, we have*

$$\sum_{t=1}^T \beta^t \le \mathbf{Regret}^{C,\alpha,\mathcal{J}_{\alpha+\epsilon}}(\mathcal{A}, Q_\alpha, T) + \sqrt{2T\ln\left(\frac{2}{\delta}\right)}$$

*Proof.* Fix the sequence $(z^t)_{t=1}^T$ and also the random coins of the algorithm, meaning $\mathcal{A}((\pi^j, z^j)_{j=1}^{t-1})$ is deterministic. Then, $(\pi^t)_{t=1}^T$ is fixed as well. By Theorem 4.4, for any sequence of environment's strategy $(z^t)_{t=1}^T$,

$$\sum_{t=1}^T \mathbb{1}\left(\mathcal{J}_{\alpha+\epsilon}\left(\bar{x}^t, \pi^t\right) \neq \emptyset\right) = \sum_{t=1}^T \text{Unfair}_{\alpha+\epsilon}(\pi^t, z^t) \leq \mathbf{Regret}^{\mathcal{L}_{C,\alpha}, \mathcal{J}_{\alpha+\epsilon}}\left(\mathcal{A}, Q_\alpha, T\right),$$

where $C = \frac{k+1}{\epsilon}$.

**Lemma B.4.**

$$\Pr_{\substack{(\bar{x}^t, \bar{y}^t)_{t=1}^T \\ \pi^t = \mathcal{A}(\cdots)}} \left[\left|\sum_{t=1}^T \mathbb{1}\left(\mathcal{J}_{\alpha+\epsilon}\left(\bar{x}'^t, \pi^t\right) \neq \emptyset\right) - \mathbb{E}_{(\bar{x}'^t, \bar{y}'^t)_{t=1}^T}\left[\sum_{t=1}^T \mathbb{1}\left(\mathcal{J}_{\alpha+\epsilon}\left(\bar{x}'^t, \pi^t\right) \neq \emptyset\right)\right]\right| \geq \gamma\right] \leq 2\exp\left(-\frac{\gamma^2}{2T}\right)$$

*Proof.* Consider the following sequence $(B^t)_{t=1}^T$:

$$B^t = \sum_{j=1}^t \mathbb{1}\left(\mathcal{J}_{\alpha+\epsilon}\left(\bar{x}^j, \pi^j\right) \neq \emptyset\right) - \mathbb{E}_{(\bar{x}'^j, \bar{x}'^j)_{j=1}^t}\left[\sum_{j=1}^t \mathbb{1}\left(\mathcal{J}_{\alpha+\epsilon}\left(\bar{x}'^j, \pi^j\right) \neq \emptyset\right)\right].$$

Note that this is a martingale: $\mathbb{E}[B^t | B^1, \dots, B^{t-1}] = B^{t-1}$ because conditioning upon previous rounds, the algorithm $\pi^t$ is deterministically chosen.

Now, we apply Azuma's inequality. Since $|B^t - B^{t-1}| \leq 1$, we have

$$\Pr\left[|B^T - B^1| \geq \gamma\right] \leq 2\exp\left(\frac{-\gamma^2}{2T}\right).$$

$\square$

**Lemma B.5.** *Fix the sequence of policies $(\pi^t)_{t=1}^T$. Say that each $\pi^t$ is $(\alpha, \beta^t)$-fair. Then:*

$$\sum_{t=1}^T \mathbb{E}_{(\bar{x}'^t, \bar{y}'^t)_{t=1}^T}\left[\mathbb{1}\left(\mathcal{J}_{\alpha+\epsilon}\left(\bar{x}'^t, \pi^t\right) \neq \emptyset\right)\right] \geq \sum_{t=1}^T \beta^t.$$

*Proof.* We will lower bound the probability of having an $\alpha'$-fairness violation on a pair of individuals among those who have arrived in a single round. Because all possible pairs within a single batch of independently arrived individuals are not independent, we resort to a weaker lower bound by dividing all of the arrivals into independent pairs. As we go through the batch, we take every two individuals to form a pair, and note that these pairs must be independent.

$$\mathbb{E}_{(\bar{x}'^t, \bar{y}'^t)_{t=1}^T}\left[\mathbb{1}\left(\mathcal{J}_{\alpha+\epsilon}\left(\bar{x}'^t, \pi^t\right) \neq \emptyset\right)\right]$$

$$= \Pr_{(\bar{x}'^t, \bar{y}'^t)_{t=1}^T}\left[\exists \tau, \tau' \in [k] : |\pi^t(x_\tau'^t) - \pi^t(x_{\tau'}'^t)| - d(x_\tau'^t, x_{\tau'}'^t) > \alpha'\right]$$

$$\geq \Pr\left[\exists i \in \left[\left\lfloor\frac{k}{2}\right\rfloor\right] : |\pi^t(x_{2i-1}'^t) - \pi^t(x_{2i}'^t)| - d(x_{2i-1}'^t, x_{2i}'^t) > \alpha'\right]$$

$$= 1 - \Pr\left[\forall i \in \left[\left\lfloor\frac{k}{2}\right\rfloor\right] : |\pi^t(x_{2i-1}'^t) - \pi^t(x_{2i}'^t)| - d(x_{2i-1}'^t, x_{2i}'^t) \leq \alpha'\right]$$

$$= 1 - \prod_{i=1}^{\lfloor\frac{k}{2}\rfloor}(1 - \beta^t)$$

$$= 1 - (1 - \beta^t)^{\lfloor\frac{k}{2}\rfloor}$$

$$\geq \beta^t$$

$\square$

Next, we combine Lemma B.4 and Lemma B.5. With probability $1 - \delta$,

$$
\mathbf{Regret}^{\mathcal{L}_{C,\alpha},\mathcal{J}_{\alpha+\epsilon}}\left(\mathcal{A}, Q_\alpha, T\right)
$$

$$
\geq \sum_{t=1}^{T} \mathbb{1}\left(\mathcal{J}_{\alpha+\epsilon}\left(\bar{x}'^{t}, \pi^{t}\right) \neq \emptyset\right)
$$

$$
\geq \sum_{t=1}^{T} \mathop{\mathbb{E}}_{(\bar{x}'^{t},\bar{y}'^{t})_{t=1}^{T}}\left[\mathbb{1}\left(\mathcal{J}_{\alpha+\epsilon}\left(\bar{x}'^{t}, \pi^{t}\right) \neq \emptyset\right)\right] - \sqrt{2T \ln\left(\frac{\delta}{2}\right)}
$$

$$
\geq \sum_{t=1}^{T} \beta^{t} - \sqrt{2T \ln\left(\frac{\delta}{2}\right)}
$$

$\square$

**Proof of Theorem 5.3** The theorem statement follows directly by combining Lemma 5.6 and Lemma B.3.

$\square$