[Reviews · NeurIPS 2020]

Review 1

Summary and Contributions: This paper discusses online batch classification problem with individual fairness constraints. It follows a metric free approach where an auditor detects fairness violation. Authors use a Lagrangian formulation to unify the constraint and the classification loss. They reduce the problem to "vanilla" online batch classification problem and proves a sublinear regret bound using CONTEXT-FTPL algorithm. Following this, they generalise the analysis to stochastic setting.

Strengths: 1. The paper solves an interesting problem which can have practical impact. 2. The paper proposes a clean reduction technique to formulate the problem and deploy online algorithms in it. 3. It removes the constraining assumptions in the existing literature such as Mahalanobis metric, while achieving the \sqrt{T} regret in loss. 4. The algorithm also achieves T^{-1/4}-fairness violations with high probability that resonates the PACF uniform convergence sample complexity.

Weaknesses: 1. The bounds have a dependency of log(|H|). Essentially, the size of the hypothesis space can be unbound or quite large. This can be a big issue to use this work in reality. 2. The reduction of the problem is not described enough in the main paper. 3. The assumptions of separators and CONTEXT-FTPL are described as passing references. But the final results depend significantly on them. 4. Being specific to previous results like which theorem or which bound is needed.

Correctness: I have not found out any error in the proves.

Clarity: The paper is clearly written but the description of the reduction from online fair batch classification to online and online batch classification, and reason behind using context FTPL should be elaborated a bit longer in the main paper.

Relation to Prior Work: 1. The paper has healthy amount of references. 2. Being specific to previous results like which theorem or which bound would be really helpful. For example, "qualitatively matches...[23]" (Remark 4.5). Here, it is easy to understand if a specific result is referred. 3. [9] depends on the dimension d whereas the results of this paper depends on log(|H|). Explaining this trade-off at least intuitively would be helpful.

Reproducibility: Yes

Additional Feedback: Except the points on clarity, prior works and weakness, please look into following points: 1. What is epsilon in Theorem 3.8 and Cor 4.4? What are the choices of it? This parameter comes from the Laplace distribution right? This should be explained. 2. An intuition behind practicality of transductive setting and separator set would be good to discuss. It is not okay to assume reader's familiarity with CONTEXT_FPL paper. 3. The generalisation in section 4 is often discussed as a stochastic setting in online learning while the previous sections are on adversarial setting. It would be good to mention that and also why the bounds for this setting improve.


Review 2

Summary and Contributions: The paper introduces a new online learning model for studying individual fairness. It provides a learning algorithm that achieves sublinear regret both for accuracy and fairness. In the case when examples are sampled from a probability distribution, the paper also gives strong generalization bounds. These results are obtained using significantly weaker assumptions than previous related papers.

Strengths: This paper addresses an important problem, and obtains significantly stronger results than prior papers. This paper's online fair learning model has much weaker assumptions and feedback format than previous similar models. The proof techniques are nontrivial. Overall, this paper provides novel and significant contributions to the field of ML fairness.

Weaknesses: The structure of the paper and the presentation of the results could be significantly improved (see details below).

Correctness: I did not find errors in the paper.

Clarity: The introduction and the problem problem statement are well written. However, the results (sections 3-4) are harder to follow and they could be presented more clearly. As an example, one of the main results of the paper, the algorithm for online fair batch classification, isn't described clearly enough in the main part of the paper; the pseudocode for it is only given in the appendix.

Relation to Prior Work: The authors do clearly discuss the paper's relationship to previous work, but much of this discussion happens in Appendix A. In my opinion, discussion of previous work is a crucial part of a paper; it shouldn't be relegated to the appendix.

Reproducibility: Yes

Additional Feedback: Response to author feedback: I thank the authors for committing to improving the presentation of the paper.


Review 3

Summary and Contributions: The paper studies online learning with individual fairness as constraints. In particular, the paper assumes the existence of an auditor that detects fairness violations rather than assuming the known similarity metric among individuals. Different from a closely related work [9] "Online learning with unknown fairness metric" by Gillen et al. NeurIPS 2018, in each round the auditor only needs to return one pair of individuals identified as fairness violation. Under this setting, the paper establishes PAC-style fairness and accuracy generalization guarantees. The main contribution is to answer the question raised in [9] and the results show that online learning under an unknown individual fairness constraint is possible even without assuming a parametric of form of the underling similarity measure.

Strengths: The presented general reduction framework, which takes any online learning algorithm as a black-box and obtains a learning algorithm that minimizes the cumulative classification error and the number of fairness violations, is sound. The removal of the previous assumptions, linear rewards and Mahalanobis distance, is also significant. It is also a nice result to see the use of the Follow-the-Perturbed-Leader approach can achieve sublinear regret with respect to both misclassification and fairness violations in the online fair batch learning setting.

Weaknesses: I do not identify any clear weakness of this work. Somehow, I would like to see how results would be different when the auditor still returns the set of all pairs of individuals with fairness violations. Similar to the work [9], the paper focuses on the fairness constraint that binds between individuals at each round. While the enforcement of the fairness constraint across rounds is difficult. But practically users may want to see to what extend the between-round individual fairness can be achieved in the proposed approaches.

Correctness: I do not check the proof details in the supplemental file. But I tend to believe the claims and method are correct.

Clarity: The paper is well written. It contains a lot of theoretical results. But the related work section should be moved (may be shortened) from the supplemental file to the main body. The rough idea of the proof technique with the use of a composition covering argument could also be discussed in the main body.

Relation to Prior Work: The related work section is included in the supplemental file. Some discussions there, especially comparison with [9] and [23], can be moved into the introduction section of the main body. But overall, relation to prior work is clearly discussed. Regarding the rebuttal, I appreciate the authors' commitment of improving the paper's presentation and clarifications of my two questions, the case of all pairs of individuals with fairness violations returned, and between-round individual fairness.

Reproducibility: Yes

Additional Feedback: It is good to consider the setting with the adaptive (and weak) fairness feedback. Some discussions, like how the approaches and results would be different under assumption of gradually change, would be helpful. I change my score from 7 to 8.

[Author Response · NeurIPS 2020]

We thank the reviewers for their effort, valuable insights, and comments!

**Presentation**   We will, as suggested by the reviewers, improve the presentation of the paper as we will:

1. Include a concise summary of the related work section in the main body.

2. Further clarify the description of the reduction from fair online batch classification to online batch classification.

3. Attempt to re-arrange and improve the presentation of the result sections (3,4) of the paper.

4. Add a description of the CONTEXT-FTPL algorithm (and also expand a bit about the transductive setting,
separator sets), and add a discussion on the motivation behind using it in our setting. We will make sure that
no prior knowledge related to CONTEXT-FTPL on the reader's side is assumed or required.

5. Attempt to be more specific when referring to prior work (exact theorems, bounds) throughout the paper.

**How results would be different when the auditor still returns the set of all pairs of individuals with fairness**
**violations (R3):** Indeed an interesting question. We have thought about it quite a bit while writing the paper - it is
still not clear to us how that can be leveraged to improve the overall guarantee. One implication of such a strong
requirement could potentially be a faster fairness convergence rate. However, for this to be done, we have to penalize
rounds diferentially according to the amount of violations, not just according to the existence of one or more violations,
as we suggest by the reduction approach in our paper. This is definitely an interesting avenue for future work, although
in terms of practicality, requiring a human auditor to point out all violating pairs might be prohibitive if the number of
individuals per round is large.

**Dependence of the bounds on** $\log(|H|)$**, comparison with the result of Gillen et al. [9], which depends on the**
**dimension of the instance space, d (R1):** It is important to stress that Gillen et al. [9] operate under a strong set of
additional assumptions, in the form of: a) Linear rewards with sub-gaussian noise, b) A metric assumption which must
be a Mahalanobis distance function, and c) The assumption that all fairness violations must be detected on every round.
It is these assumptions that, in turn, allow them to achieve bounds that depend on the dimension of the instance space,
partly due to the fact that algorithms for this problem absent fairness constraints that achieve such a dependence are
well-known. They indeed utilize a form of the LinUCB algorithm. Our setting, however, removes all three mentioned
assumptions, leaving us in the more difficult, non-parametric case - for which no algorithms with dependence on the
dimension of the instances are known. Also, note that $\log(|H|)$ term in our bounds stems directly from the regret
guarantee of CONTEXT-FTPL, while any other algorithm in the adversarial setting can be used as a blackbox for our
problem; it's just that we don't know of any other algorithm that can achieve better guarantees than CONTEXT-FTPL
in terms of the complexity of the hypothesis class without additional assumptions.

In addition, given that we operate in a non-parametric, adversarial setting, we cannot even hope for a mistake bound
which depends on the VC-Dimension of $H$—there exist simple classes $H$ with bounded VC dimensions (e.g., 1-
dimensional thresholds) for which sub-linear regret bounds are not possible with adversarial contexts. As an interesting
future direction, it would be interesting to see if, when operating in the stochastic arrivals setting (as in section 4),
the fair online batch problem can be reduced to the (stochastic) online batch setting. Such a reduction would allow,
for example, to incorporate efficient algorithms for the stochastic online batch setting, which would then replace
the dependence on $\log(|H|)$ by the VC-dimension of $H$. One obvious hurdle stems from the fact that even though
individual instances arrive stochastically in this setting, the auditor is still allowed to select an arbitrary violating pair on
every round adaptively. Replacing the $\log(|H|)$ dependence is therefore non-trivial in our setting, and we consider it a
challenging and intriguing question for future work.

**The** $\epsilon$ **parameter (R1):** This is a slack parameter, representing the sensitivity of the auditor. Due to the nature of the
adversary that can choose very similar instances and charge a pair whose fairness violation is infinitesimally bigger than
the allowed threshold $\alpha$, linear fairness regret seems unavoidable without the slack. Thus, in our model, the auditor
reports fairness violations of size at least $\alpha + \epsilon$. As shown in the regret guarantee Corollary 3.6 and Theorem 3.8, we
characterize the trade-off between the slack allowed and the actual regret for fairness and accuracy.

**Practically, to what extent the between-round individual fairness can be achieved in the proposed approaches**
**(R3):** We note that enforcing individual fairness across rounds is challenging with existing impossibility results from
Gupta and Kamble [4]. Their results show that in the adversarial arrival setting, enforcing individual fairness across
rounds would imply linear regret even when the fairness metric is known: linear regret is unavoidable if the learner has
to treat even the future instances as similar as the past instances that were misclassified. However, in the stochastic
arrivals setting, our fairness generalization result does imply it is possible to achieve approximate individual fairness
across rounds.

**The rough idea of the composition covering argument could also be discussed in the main body (R3):** Given
space constraints, we have made an attempt to convey the core idea in the "high-level proof idea" of lemma 4.6.

[Meta-Review · NeurIPS 2020]

The paper concerns a new online learning problem subject to the constraint of individual fairness. It provides a framework that reduces online classification in the considered model to standard online classification, obtaining an algorithm with sublinear regret both in terms of accuracy and fairness, as well as strong generalization bounds in the i.i.d. case. All the reviewers liked the paper and the proposed metric-free approach. The appreciated an interesting problem formulation and a clean reduction technique to a known online learning problem. The paper received very high uniform scores of 8 from each reviewer. The reviewers found some issues with the presentation, and I hope the authors will address them in the final version of the manuscript.